# Analysis of behavioral flow resolves latent phenotypes

Lukas M. von Ziegler [1,2,5], Fabienne K. Roessler [1,2,5], Oliver Sturman [1,2,3], Rebecca Waag [1,2], Mattia Privitera [1,2], Sian N. Duss [1,2], Eoin C. O'Connor [4] & Johannes Bohacek [1,2,3]

The accurate detection and quantification of rodent behavior forms a cornerstone of basic biomedical research. Current data-driven approaches, which segment free exploratory behavior into clusters, suffer from low statistical power due to multiple testing, exhibit poor transferability across experiments and fail to exploit the rich behavioral profiles of individual animals. Here we introduce a pipeline to capture each animal's behavioral flow, yielding a single metric based on all observed transitions between clusters. By stabilizing these clusters through machine learning, we ensure data transferability, while dimensionality reduction techniques facilitate detailed analysis of individual animals. We provide a large dataset of 771 behavior recordings of freely moving mice—including stress exposures, pharmacological and brain circuit interventions—to identify hidden treatment effects, reveal subtle variations on the level of individual animals and detect brain processes underlying specific interventions. Our pipeline, compatible with popular clustering methods, substantially enhances statistical power and enables predictions of an animal's future behavior.

The reliable detection of complex mouse behavior in biomedical research settings is critical to gain insights into disease conditions, genetic phenotypes and internal states (that is, emotion, motivation and so on). Pioneering work leveraged video-based body-center tracking and data mining in the open field test (OFT)—routinely used to capture unconstrained rodent behavior—to show that the behavioral repertoire in simple test setups contains rich information about an animal's traits[1,2]. The advent of pose-estimation technology[3–5] has renewed interest in analyzing animal behavior using data-driven methods that operate independently of human intervention[6–8]. Based on raw or processed video data, these approaches segment behavior using clustering algorithms or more sophisticated state-space models and have revealed a remarkable complexity underlying even the simplest of behavioral tests[9–14].

Despite this rapid progress, such advancements have given rise to four major challenges, which we address in this work. (1) Data-driven analysis of animal behavior promises to better resolve differences between experimental groups, by analyzing large numbers of behavioral variables[6–8]. However, this increases statistical demands from multiple testing corrections, thus decreasing the power to detect group differences. We solve this by introducing a 'behavioral flow analysis' (BFA), which uses a single metric to identify treatment effects based on the observed transitions between the different behavioral clusters. (2) Clusters that represent behaviors vary between experiments, making it difficult to compare clustering results between different experiments. We address this using a large dataset to train a supervised machine learning classifier to recognize established behavioral clusters in newly encountered datasets. (3) It remains challenging to compare behavioral results across large sets of experiments. We combine BFA and stabilized clusters with dimensionality reduction techniques to generate a single high-dimensional data point for each animal. This 'behavioral flow fingerprinting' (BFF) approach allows large-scale comparisons of animal behavior across a wide range of

[1]Laboratory of Molecular and Behavioral Neuroscience, Institute for Neuroscience, Department of Health Sciences and Technology, ETH Zurich, Zurich, Switzerland. [2]Neuroscience Center Zurich, ETH Zurich and University of Zurich, Zurich, Switzerland. [3]ETH 3R Hub, ETH Zurich, Zurich, Switzerland. [4]Roche Pharma Research and Early Development, Neuroscience and Rare Diseases, Roche Innovation Center Basel, Basel, Switzerland. [5]These authors contributed equally: Lukas M. von Ziegler, Fabienne K. Roessler. ✉e-mail: johannes.bohacek@hest.ethz.ch

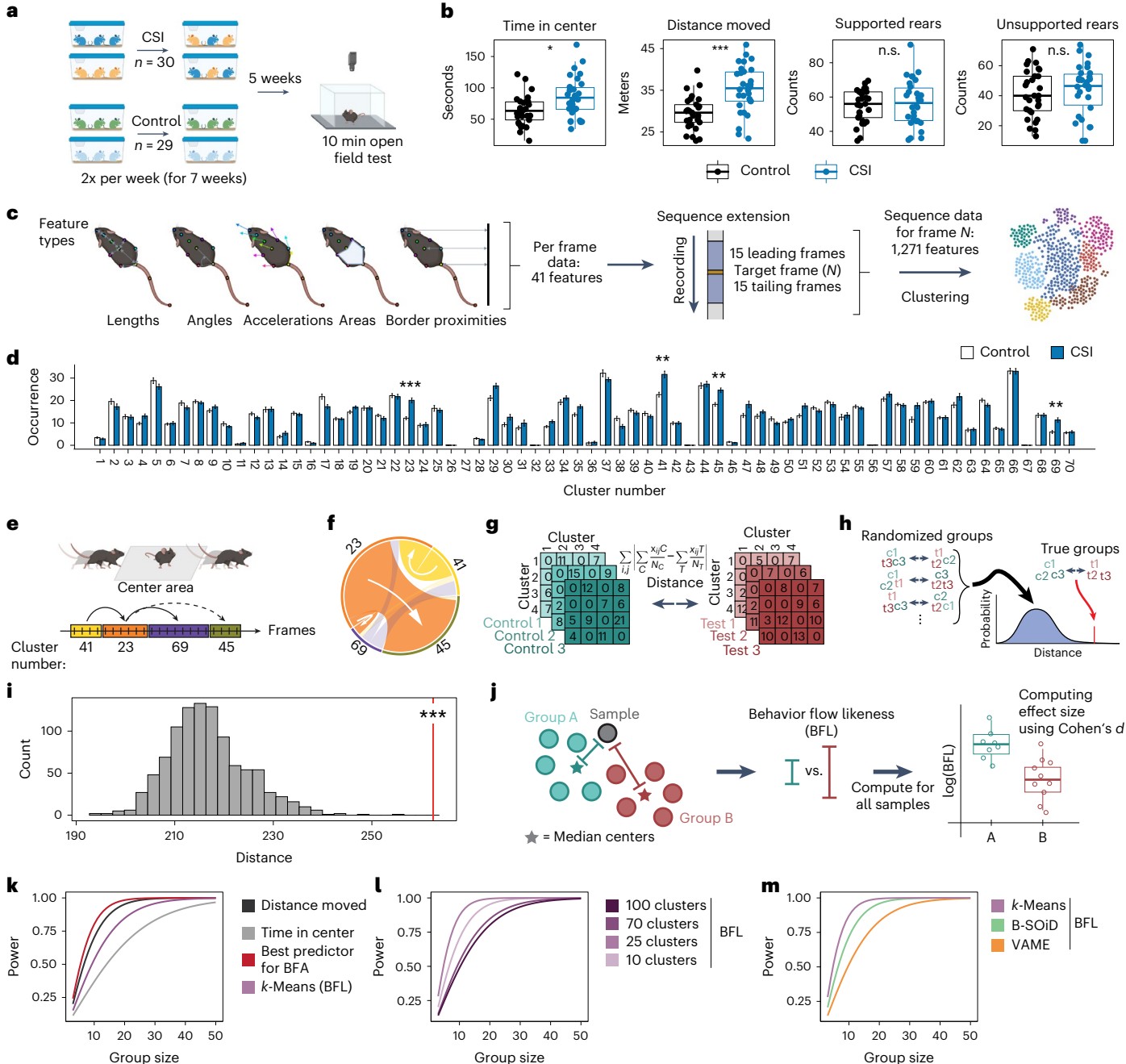

**Fig. 1 | BFA increases power to detect phenotypes. a**, A schematic showing the experimental design for CSI. **b**, Classical behavior readouts in the OFT show that CSI mice (*n* = 30) spend more time in the center (two-tailed *t*-test, *t*(53) = 2.96, adjusted *P* = 4.47 × 10⁻²) and travel greater distance (two-tailed *t*-test, *t*(52) = 4.55, adjusted *P* = 7.08 × 10⁻⁴) than controls (*n* = 29). **c**, Feature extraction based on pose-estimation tracking and sequential feature integration for subsequent clustering. **d**, The *k*-means cluster occurrence in CSI (*n* = 30, controls: *n* = 29; two-tailed *t*-tests with multiple testing correction). **e**, A schematic example of behavioral flow based on cluster transitions. **f**, The average behavioral flow over all animals between example clusters. The white arrows display the direction of transition. **g**, Schematic of computing Manhattan distance to compare behavioral transition matrices between groups. **h**, The permutation approach used for BFA

to compare the transition distance based on the true group assignment versus the randomized group assignment. c, control; t, test. **i**, BFA reveals a treatment effect for CSI (one-tailed *z*-test, percentile 99.9, *z* = 5.72, *P* = 5.28 × 10⁻⁹, *d* = 0.97). **j**, Schematic of computing the BFL score to estimate effect sizes. **k**, Power analysis comparing classical readouts ('distance moved' and 'time in center') with analysis of cluster transitions. **l**, The number of clusters influences power to detect treatment effects. **m**, Power analysis comparing three different clustering algorithms. *P* values and adjusted *P* values are denoted as *<0.05, **<0.01 and ***<0.001; n.s., not significant. For box plots, the center line denotes the median value, while the bounding box delineates the 25th to 75th percentiles. The whiskers represent 1.5 times the interquartile range from the lower and upper bounds of the box. The error bars in the bar plot denote mean ± s.e.m.

experimental manipulations. (4) It has not yet been demonstrated that behavioral personality profiling—akin to a detailed clinical assessment in humans—is possible in mice. We introduce an approach that compares the similarity of each animal's behavioral flow to the median group profiles, thus deriving a 'behavioral flow likeness' (BFL) score,

which generates a behavioral profile that allows behavioral predictions in future test settings.

Using a series of stress paradigms, and stress-related pharmacological and circuit neuroscience interventions, we test and validate our approach across many behavioral datasets, across different types

of tests and in independent laboratories. An implementation of the analysis pipeline is freely available to others through the BehaviorFlow package (https://github.com/ETHZ-INS/BehaviorFlow). Our results are in line with the reduce-and-refine principles set forth by animal welfare regulations, as our approach increases statistical power, reduces the number of animals required for experiments and increases the information extracted from each experimental animal.

## Results

### BFA reveals treatment effects

To test whether an unsupervised approach would reveal known phenotypes, we first turned to a published dataset[15] in which mice were exposed to chronic social instability (CSI) stress or to control handling, before being tested on the OFT (Fig. 1a). Classical behavior analysis—as described previously[15,16]—showed that CSI mice spent more time in the center of the open field and traveled greater distances, while rearing behavior was not affected (Fig. 1b).

To test how well an unsupervised clustering approach performs at resolving group differences, we tracked 13 body points using the pose-estimation tool DeepLabCut[3,17], transformed the tracking data into a set of 41 features and resolved them over a sliding time window (±15 frames) to describe short temporally resolved sequences (Fig. 1c). The resulting data contained 1,271 dimensions for each frame. We then used a computationally efficient $k$-means clustering algorithm to resolve different behaviors. To determine the best number of clusters, we chose an approach previously used for similar datasets[10,14] by first partitioning the recorded behavior into 100 clusters and then choosing the number of clusters that represented 95% of the imaging frames. This mark indicated about 70 clusters (Extended Data Fig. 1a), so we subsequently reran the clustering approach using 70 clusters. Although nominal $P$ values revealed that CSI and control animals behaved differently on many of these clusters, only 4 out of 70 clusters survived the appropriate multiple testing correction (Benjamini–Yekutieli; Fig. 1d and Supplementary Fig. 1a). Visual inspection of these significant clusters (two-tailed $t$-tests, adjusted $P < 0.05$) reveals—in agreement with the classical analysis—that they capture the time that animals spend in the center of the open field. Specifically, these four clusters represent movement of the mouse from the periphery into the center (cluster 41; Supplementary Video 1), movement in or through the center (cluster 23; Supplementary Video 2), orienting and turning in the center (cluster 69; Supplementary Video 3) and movement from the center back to the periphery (cluster 45; Fig. 1e and Supplementary Video 4).

Because behavior emerges as a dynamic property of moment-to-moment action, we took advantage of the frame-by-frame resolution of the clustering data to assess behavioral flow (that is, the temporal sequence in which one cluster transitions to another cluster) in each animal (Fig. 1e). Analyzing behavioral flows across all animals (independent of group assignment) demonstrated that many clusters had ingoing and outgoing transitions that were much more likely to occur (Extended Data Fig. 1b). An example is the clusters identified as significant (two-tailed $t$-tests, adjusted $P < 0.05$) between CSI and control mice, where behavioral flow indicates that, when mice move from the periphery to the center (cluster 41), this can be followed by center exploration (cluster 69) or movement through the center (cluster 23), which is most often followed by a movement back into the periphery (cluster 45; Fig. 1e,f). However, when we tested for group differences using cluster transitions, none of them remained significant (two-tailed $t$-tests, adjusted $P < 0.05$) after multiple testing correction (1,753 observed transitions out of 4,830 possible transitions). To demonstrate that the information gain is punished by multiple testing correction, we designed a sensitivity assay (Extended Data Fig. 1c). When using unadjusted $P$ values, both cluster usage and transition occurrences perform better than classical analyses (Extended Data Fig. 1d). However, when applying multiple testing corrections, both cluster and transition measures perform poorer than classical analyses in detecting phenotype differences (Extended Data Fig. 1e).

To address this multiple-testing-problem, we used a statistical approach to detect group differences based on the combined behavioral flow data, which we term behavioral flow analysis (BFA). The BFA method first defines the difference between the two experimental groups based on the Manhattan distance between group means across all behavioral transitions (Fig. 1g). To assess if this distance is significantly larger than expected, we used a permutation approach where randomized group assignments were generated using the original data to estimate a null distribution of the intergroup distance (Fig. 1h). Then, we calculated the percentile and tested the true distance against the null distribution using a right-tailed $z$-test, which revealed a strong group difference that is very unlikely to be due to chance (Fig. 1i). To rule out that BFA arbitrarily generates effects, we randomly divided only control animals and only CSI animals into two subgroups ($n = 14$–$15$ per group) and showed that BFA does not detect differences between the two control groups, nor between the two CSI groups (Extended Data Fig. 1f).

Can analyzing the entire behavioral flow increase statistical power? Power analyses are traditionally calculated on the basis of a single behavioral measure. To estimate the effect size between the high-dimensional behavioral flow profiles of two groups, we developed a behavioral flow likeness (BFL) score, to compare each animal's behavioral profile to the median of the two groups. Based on these BFL scores, we computed the effect size of the CSI treatment using Cohen's $d$ (Fig. 1j). For CSI, which strongly increases locomotion, traditional power calculation shows that 'distance moved' yields more power than 'time in center' (Fig. 1k). Power analysis using the BFL-based approach yields more power than 'time in center', but less than 'distance moved' (Fig. 1k). However, picking the single transition with the lowest adjusted $P$ value (termed 'best predictor for BFA') yielded a higher power compared with 'distance moved' (Fig. 1k). Thus, the BFL approach offers an unbiased estimation of effect sizes and enables power calculations based on the entire behavioral profile of an animal. Systematically changing the temporal integration period (from ±5 frames to ±30 frames) showed only minor differences in power, but our initial choice of ±15 frames performed best (Extended Data Fig. 1g). In contrast, a systematic comparison of cluster numbers (from 10 to 100) revealed that BFL based on 25 clusters yields the highest power (Fig. 1l). Similarly, 25 clusters yielded by far the largest statistical power when using sensitivity assays based on $P$ values, where group sizes were successively reduced in silico (Extended Data Fig. 1h). Finally, we demonstrated that our BFA and effect size estimation also work with the output of other clustering algorithms used for analyzing rodent behavior, the software packages VAME[14] and B-SOiD[9] (Supplementary Note 1).

### Cluster stabilization enables comparisons across experiments

To sample behavior from mice exposed to various stress-related challenges, we added two new behavioral datasets: (1) an experiment in which mice were tested 45 min and 24 h after a short swim stress exposure (Fig. 2a) and (2) a pharmacological stress model that allowed us to introduce a graded response by injecting mice with escalating doses of yohimbine (Fig. 2b). Yohimbine is an α2-adrenergic receptor antagonist that triggers noradrenaline release by disinhibiting the locus coeruleus[18,19]. To obtain comparable clustering results across these experiments, all the data would need to be used in one big clustering experiment, which can become computationally expensive. Further, although $k$-means allows calculating which cluster center is closest to a new data point, other clustering algorithms (for instance, density-based clustering as applied by B-SOiD) would need access to the original clustering data to embed new behavioral recordings. To offer a computationally efficient solution for every choice of clustering algorithm, we stabilized clusters using a classifier-in-the-middle approach. To this end, we selected a random set of 60 animals from

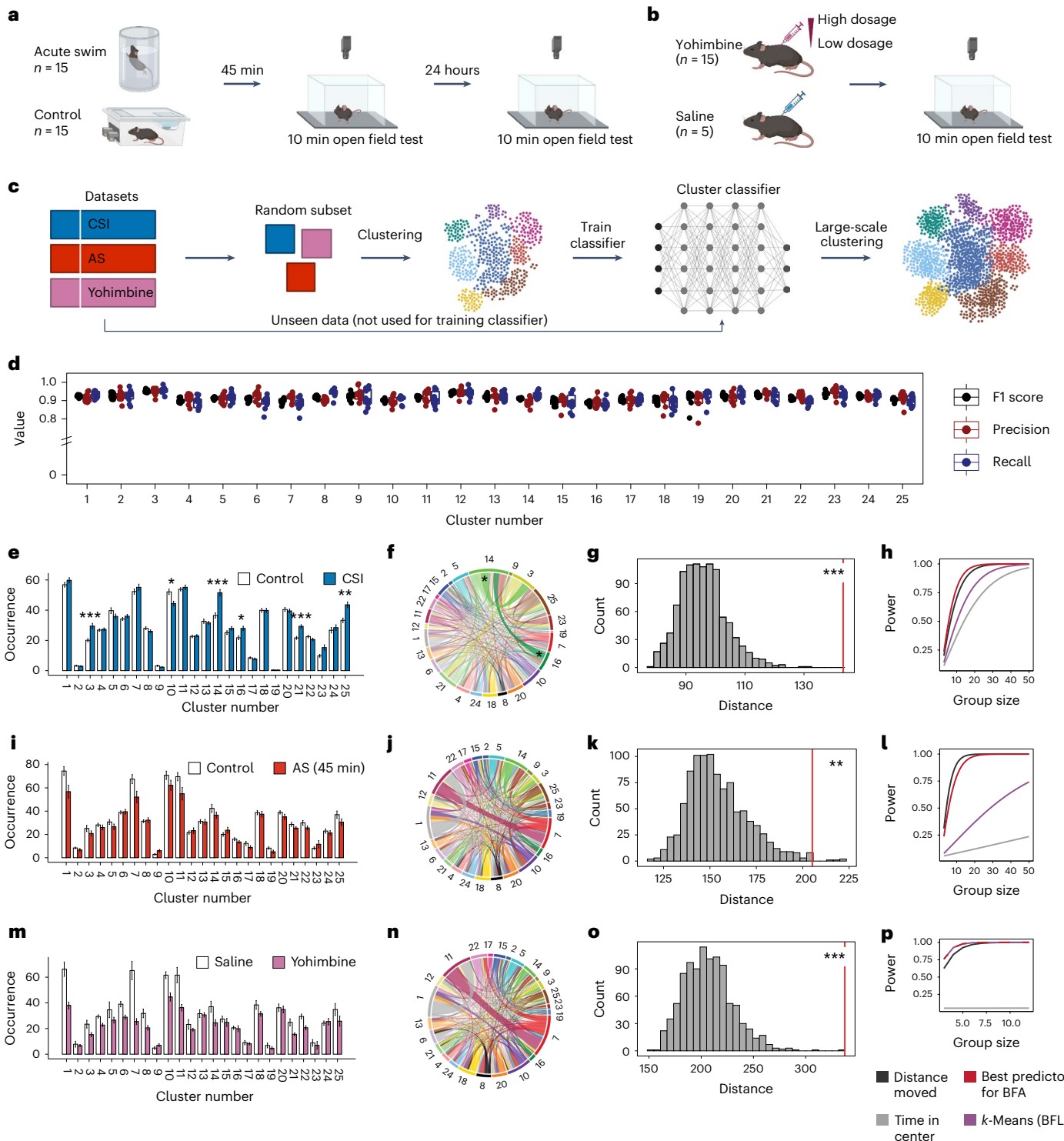

**Fig. 2 | Cluster stabilization enables comparisons across experiments.**
**a**,**b**, A schematic of the experimental design for OFT after acute swim stress (AS) (**a**) or after yohimbine injections (**b**). **c**, Clustering across large datasets. **d**, Classifier performance on 10-fold cross-validation for each cluster. **e**, Quantification of cluster occurrences in CSI ($n = 30$, controls: $n = 29$; two-tailed $t$-tests with multiple testing correction). **f**, Absolute differences in behavioral flow in control versus CSI. For each cluster, the absolute difference in the observed number of transitions between groups is plotted. **g**, BFA reveals a treatment effect for CSI (one-tailed $z$-test, percentile 99.9, $z = 6.02$, $P = 8.63 \times 10^{-10}$, $d = 0.92$). **h**, Power analysis in CSI. **i**, Cluster occurrences in AS (45 min) ($n = 15$, controls: $n = 15$; two-tailed $t$-tests with multiple testing

correction). **j**, The absolute difference in behavioral flow in control versus AS (45 min). **k**, BFA reveals a treatment effect for AS at 45 min (one-tailed $z$-test, percentile 99.4, $z = 3.09$, $P = 1.01 \times 10^{-3}$, $d = 0.53$). **l**, Power analysis in AS. **m**, Cluster occurrences in yohimbine ($n = 15$, controls: $n = 5$; two-tailed $t$-tests with multiple testing correction). **n**, The absolute difference in behavioral flow in saline versus yohimbine. **o**, BFA reveals a treatment effect for yohimbine (one-tailed $z$-test, percentile 99.8, $z = 5.56$, $P = 1.38 \times 10^{-8}$, $d = 2.91$). **p**, Analysis of cluster transitions shows higher power in detecting treatment effects for yohimbine. $P$ values and adjusted $P$ values are denoted as *<0.05, **<0.01 and ***<0.001. The error bars in the bar plots denote mean ± s.e.m.

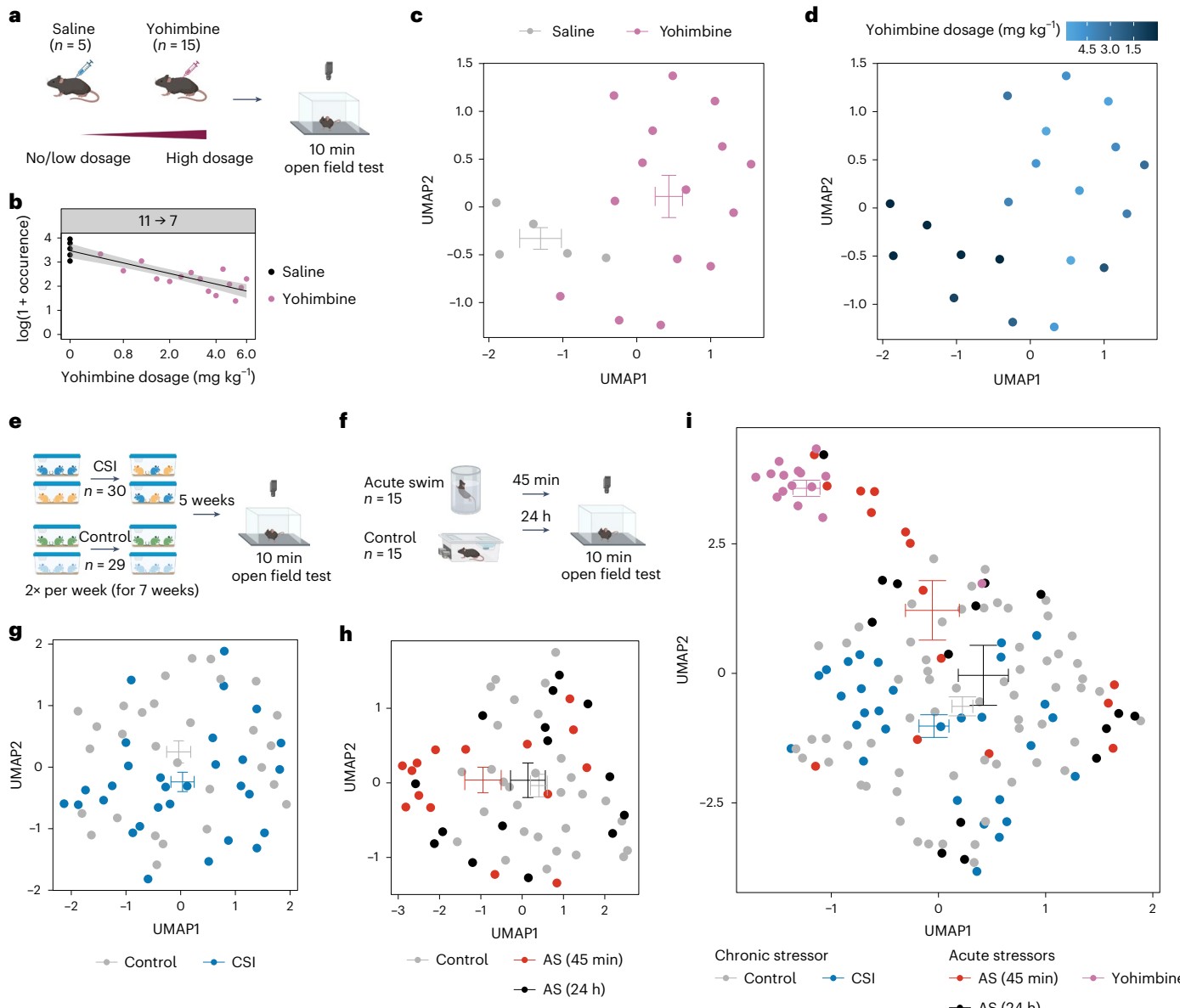

**Fig. 3 | BFF captures individual differences in high-dimensional space.**
**a**, A schematic of the experimental design showing the escalating dose of
yohimbine. **b**, Using dose in a log-linear model reveals one significant transition
(linear regression, $R^2 = 0.75$, $F(1,18) = 55.44$, adjusted $P = 1.64 \times 10^{-3}$). The gray
band shows the 95% confidence interval. **c**, BFF reveals a separation between
mice treated with vehicle ($n = 5$) versus yohimbine ($n = 15$) when applying UMAP
dimension reduction to their transition matrices. **d**, BFF can also visualize the
drug dose delivered to each animal. **e**, A schematic of the experimental design of

the CSI experiment. **g**, BFF applied to the CSI ($n = 30$, controls: $n = 29$) dataset.
**f**, A schematic of the experimental design of the acute swim (AS) stress
experiment. **h**, BFF applied to the acute swim stress ($n = 15$ at each time point),
controls: $n = 15$) dataset. **i**, Plotting BFF embeddings across all three experiments
(CSI: $n = 30$, AS: $n = 15$ (at each time point), yohimbine: $n = 15$, controls
(combined): $n = 49$) reveals a separation of all experimental groups in 2D space.
For every UMAP embedding, the crossbars represent the average UMAP1 and
UMAP2 values with s.e.m. for each group.

all behavioral experiments (that is, 20 animals each from CSI, acute
swim and yohimbine) for performing $k$-means clustering to generate
25 clusters, as this number of clusters was sufficient to detect phe-
notypical differences as shown above (Fig. 1k). Afterward, we used
these clustering results to train a neural network that can imitate the
clustering on the random subset for the rest of the data (Fig. 2c). Using
a 10-fold cross-validation, we found that this approach had good (>0.9)
precision and recall values across all 25 clusters (Fig. 2d).

Using this cluster stabilization approach, we first reanalyzed the
CSI experiment. As we now only used 25 clusters, multiple testing
correction was less punishing and we identified 6 significant clusters
(two-tailed $t$-tests, adjusted $P < 0.05$; Fig. 2e and Supplementary Fig. 1b)
and two significant transitions (two-tailed $t$-tests, adjusted $P < 0.05$;

Fig. 2f). Visual inspection revealed—consistent with the original analy-
sis—that all clusters that occurred more frequently in CSI mice captured
behaviors related to active locomotion in the center: movement in
the center of the open field (cluster 3; Supplementary Video 5), the
initiation of movement from the periphery to the center (cluster 14;
Supplementary Video 6), fast locomotion crossing the center or mov-
ing from center to periphery (cluster 21; Supplementary Video 7) and
exploration/turning in the center or movement toward the periphery
(cluster 25; Supplementary Video 8). In contrast, the only underrep-
resented cluster in CSI mice was movement or exploration along the
periphery of the open field (cluster 10; Supplementary Video 9). For
a full description of all clusters, see Supplementary Table 1. The BFA
pipeline reproduced the strong phenotype (Fig. 2g), and the power was

increased (Fig. 2h). Taken together, our approach captures meaningful behavioral motifs and interpretable transitions, independent of the number of clusters, and also when the original dataset is reanalyzed using a set of clusters derived from training on subsets of videos from various experiments.

We next applied the same analysis pipeline using the clustering classifier to open field behavior assessed 45 min after an acute swim stress. None of the 25 clusters and none of the observed transitions revealed any group differences (Fig. 2i,j and Supplementary Fig. 1c), yet BFA readily identified a significant group difference (one-tailed $z$-test, $P = 1.01 \times 10^{-3}$; Fig. 2k). When we instead analyzed open field behavior 24 h after swim stress, neither the 25 clusters or their transitions nor BFA revealed a discernible phenotype (Extended Data Fig. 2a–c), consistent with our previous observations that swim stress induces only transient changes in mouse behavior[20]. These results suggest that BFA can detect group differences when other methods fail, but it does not create arbitrary effects in a scenario when no differences would be expected.

Next, we applied the same analysis pipeline to mice injected with escalating doses of yohimbine. For this, we grouped all yohimbine-injected mice into one group (yohimbine, $n = 15$), and compared them with the five saline-injected controls (Fig. 2b). Because of the small number of mice in the control group, nominally large effects on various individual clusters did not yield statistical significance after multiple testing correction (two-tailed $t$-tests, adjusted $P < 0.05$; Fig. 2m and Supplementary Fig. 1d), nor on the level of observed transitions (Fig. 2n). However, BFA was again able to reveal a significant group difference (one-tailed $z$-test, $P = 1.38 \times 10^{-8}$, Fig. 2o), showcasing the power of this analysis even when applied to experiments with intentionally high variability and small group sizes. In line with our previous findings that yohimbine reduces supported rearing[19], yohimbine strongly suppresses clusters 1 and 11, which capture the initiation and termination of a supported rear, respectively (Supplementary Videos 10 and 11). Power analyses for both acute swim test (Fig. 2l) and yohimbine (Fig. 2p) reveal that the metric 'best predictor for BFA' yields similar or higher power for detecting group differences compared with standard OFT readouts.

## 2D embedding captures individual differences

Beyond identifying group differences, it is necessary to understand behavior at the level of individual animals. Unsupervised approaches can resolve differences between different drugs and drug doses[11]. We thus asked whether behavioral flow would be sensitive to drug dosage in the yohimbine experiment (Fig. 3a). Modeling occurrence of transitions as a function of dosage showed that the transition from a supported rear (cluster 11; Supplementary Video 11) to the subsequent turning motion and onset of movement (cluster 7; Supplementary Video 12) occurred less frequently at higher yohimbine doses (Fig. 3b). To further explore individual differences based on behavioral flow, we considered that the behavioral flow diagrams represent high-dimensional data for each animal (625 possible transitions between 25 clusters). We applied Uniform Manifold Approximation and Projection (UMAP) to project the high-dimensional behavioral transition matrix of each animal onto a single data point in two-dimensional (2D) space. This clearly separated saline from yohimbine groups (Fig. 3c), and resolved low versus high dosages, despite the small animal numbers used in this experiment (Fig. 3d). As this approach provides a unique description of each animal's behavioral repertoire, we refer to it as behavioral flow fingerprinting (BFF).

Being able to stabilize the unsupervised clustering analyses across experiments, and to capture individual behavioral response profiles of each animal, we explored whether we could plot individual behavioral profiles across experiments. First, individual BFF embeddings revealed that stress groups shifted away from controls after CSI (Fig. 3e,g) and 45 min (but not 24 h) after swim stress exposure (Fig. 3f,h). However, in line with the large behavioral variance observed in these experiments,

the embeddings also show the large overlap between groups. Next, we normalized behavioral flow to the internal control group within each dataset and plotted the BFF embedding across all three experiments (Fig. 3i). This is in notable contrast to a previous attempt, where each group was compared not with an internal control, but with all other groups combined[11], thus potentially overestimating the power to identify effects. The resulting 2D embedding shows that yohimbine—which triggers noradrenaline release—induces distinct changes that separate the yohimbine group strongly from control animals. Acute stress—during which noradrenaline is released as well—shifts the animals toward the yohimbine group (45 min), an effect that disappears as stress effects subside (24 h). This shows that combining BFF with cluster stabilization allows visualizing treatment effects and comparing the impact of various manipulations across experiments and individuals in ways that were previously not possible.

## Cluster transfer to new datasets and data integration

Next, we tested how well the analysis pipeline can be transferred to two different experiments that were not used to perform clustering and train the clustering classifier. In the first experiment, mice were exposed to chronic restraint stress (CRS) for 90 min per day on 10 consecutive days (Fig. 4a) and tested in the OFT 45 min after the last stress exposure. In the second experiment, we triggered noradrenaline release directly in the brain using chemogenetic (designer receptor exclusively activated by designer drugs, DREADD, hM3Dq) activation of the locus coeruleus[21,22] (Fig. 4b). We recorded open field behavior directly after DREADD activation. Then, we performed clustering as described above using the cluster classifier trained on the CSI, swim stress and yohimbine data (Fig. 4c).

We found high consistency in behavioral flow between control animals in the original datasets when contrasted with control animals in the new datasets (Fig. 4d). Further, we observed that clusters 1 and 11 (Supplementary Videos 13 and 14) mapped to onset and offset of supported rearing, respectively, consistent with the previous experiments (Supplementary Videos 10 and 11; for a description of all clusters, see Supplementary Table 1). This demonstrates a reproducible clustering transfer for new datasets. Further analysis revealed six significant clusters (two-tailed $t$-tests, adjusted $P < 0.05$) in the CRS experiment (Fig. 4e and Supplementary Fig. 1e), and while no cluster transition survived multiple testing correction (Fig. 4f), BFA identified a significant group effect (one-tailed $z$-test, $P = 3.18 \times 10^{-8}$; Fig. 4g). In the DREADD experiment, no significant clusters or transitions were detected (two-tailed $t$-tests, adjusted $P < 0.05$; Fig. 4i,j and Supplementary Fig. 1f), yet BFA was again able to reveal a significant group effect (one-tailed $z$-test, $P = 1.81 \times 10^{-3}$; Fig. 4k). In both the CRS and DREADD experiments, our pipeline increased power to detect group effects compared with classical behavioral readouts (Fig. 4h,l). Visually, BFF clearly separated phenotypes in the 2D embedding in both datasets (Fig. 4m,n).

We then plotted all experiments presented thus far in one 2D embedding (Fig. 4o). All manipulations that acutely trigger noradrenaline release (acute swim, yohimbine and DREADD) were shifted away from the control groups in the same direction. In sharp contrast, the two chronic stressors (CSI and CRS) induced distinct phenotypes, with animals being shifted toward two different directions. This demonstrates that our analysis pipeline can be employed on different datasets that were not used for training the clustering classifier and that, across experiments, this approach can describe behavioral response profiles in ways that are consistent with the underlying brain processes. Similar results could be obtained when our pipeline was applied to a different clustering approach (VAME; Supplementary Note 2).

## Quantifying individual variability to predict behavior

In mice exposed to swim stress we noted—despite the clear treatment effect—that several animals embedded closer to controls (Fig. 4o), raising the possibility that they might have been less responsive to the

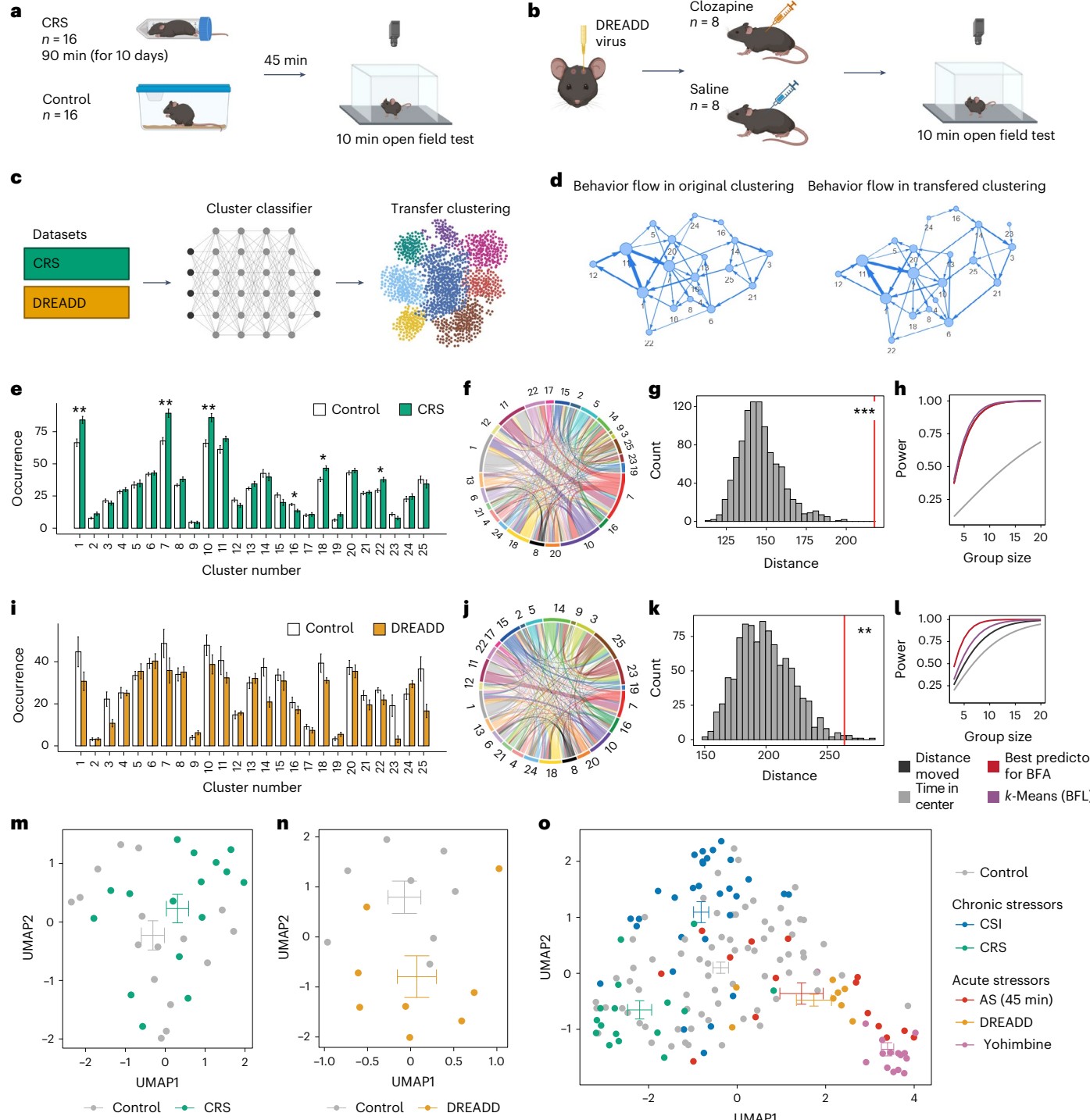

**Fig. 4 | Clustering is transferable to new datasets with the same experimental setup. a,b**, A schematic of the experimental design for OFT after CRS (**a**) or after DREADD activation of the locus coeruleus (**b**). **c**, Cluster transfer to new datasets that were not used for the initial clustering. **d**, Comparison of average behavioral flow in control animals reveals a similar pattern between original clustering (left; CSI, acute swim stress (AS) and yohimbine) and transferred clustering (right; CRS and DREADD). Only transitions with an average appearance >5 are shown. **e**, Quantification of cluster occurrences in CRS ($n = 16$, controls: $n = 16$; two-tailed $t$-tests with multiple testing correction). **f**, The absolute differences in behavioral flow in control versus CRS. **g**, BFA reveals a treatment effect for CRS (one-tailed $z$-test, percentile 99.9, $z = 5.41$, $P = 3.18 \times 10^{-8}$, $d = 1.81$). **h**, Power analysis for the CRS experiment. **i**, Quantification of cluster occurrences after DREADD

activation of the locus coeruleus (DREADD: $n = 8$, controls: $n = 8$; two-tailed $t$-tests with multiple testing correction). **j**, Absolute differences in behavioral flow in saline versus clozapine. **k**, BFA reveals a treatment effect for the DREADD experiment (one-tailed $z$-test, percentile 99.2, $z = 2.91$, $P = 1.81 \times 10^{-3}$, $d = 1.58$). **l**, Power analysis for the DREADD experiment. **m,n**, BFF using dimensionality reduction for the CRS experiment (CRS: $n = 16$, controls: $n = 16$) (**m**) and for the DREADD experiment (DREADD: $n = 8$, controls: $n = 8$) (**n**). **o**, BFF embeddings across original (CSI: $n = 30$, AS: $n = 15$, yohimbine: $n = 15$) and new (CRS: $n = 16$, DREADD: $n = 8$, controls (combined): $n = 73$) experiments. $P$ values and adjusted $P$ values are denoted as *<0.05, **<0.01 and ***<0.001. The error bars in the bar plots denote mean ± s.e.m. For every UMAP embedding, the crossbars represent the average UMAP1 and UMAP2 values with s.e.m. for each group.

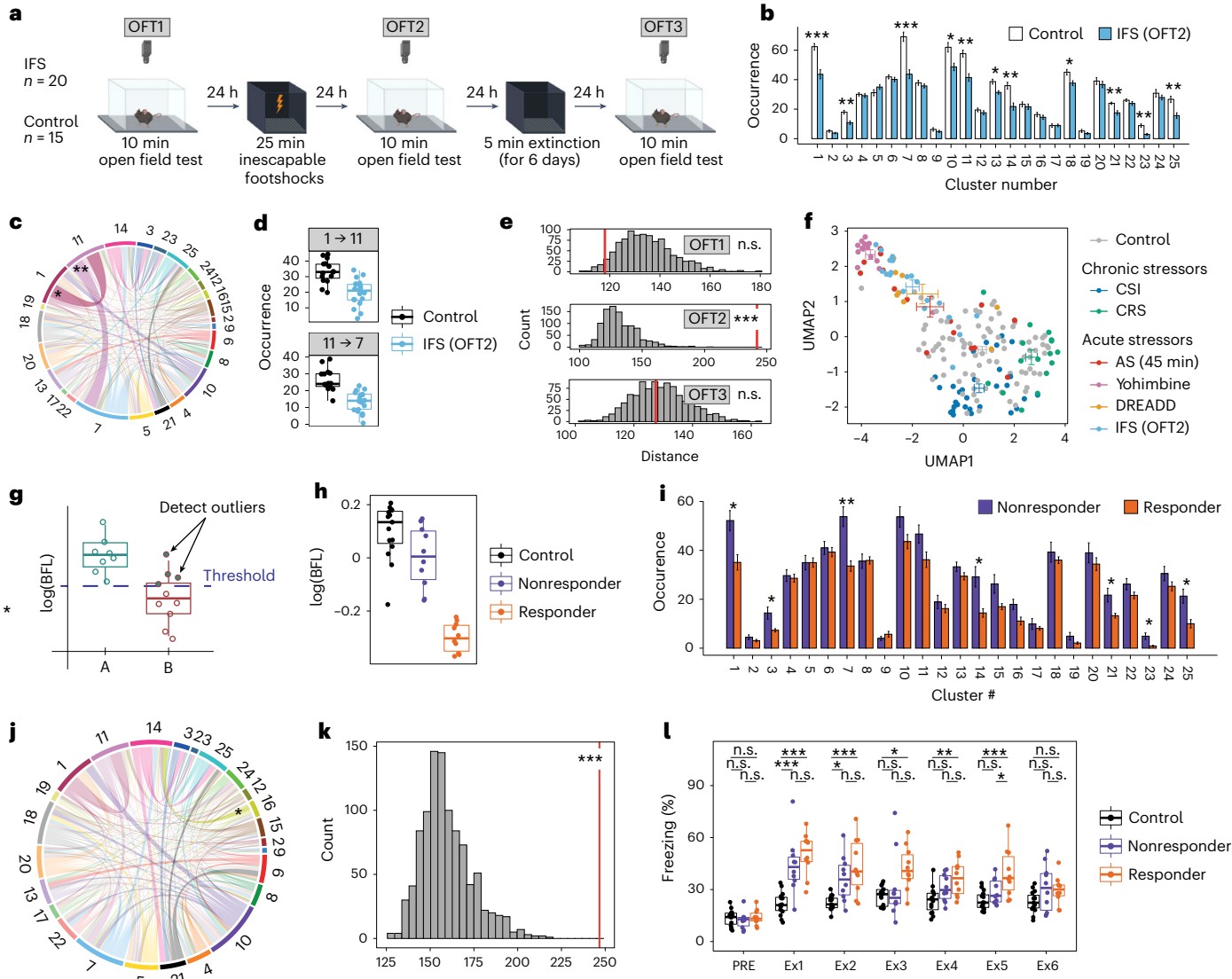

**Fig. 5 | BFF captures individual variability and allows behavioral predictions.** **a**, A schematic of the experimental design for IFS. **b**, Cluster occurrences in IFS (OFT2: $n = 20$, controls: $n = 15$; two-tailed $t$-tests with multiple testing correction). **c**, The absolute difference in behavioral flow in control versus IFS (OFT2). **d**, Significant transition occurrences in control ($n = 15$) versus IFS (OFT2, $n = 20$). **e**, BFA reveals a treatment effect of IFS only during OFT2 (one-tailed $z$-test, OFT1: percentile 3.9, $z = -1.54$, $P = 9.39 \times 10^{-1}$; OFT2: percentile 99.9, $z = 7.77$, $P = 3.89 \times 10^{-15}$, $d = 1.63$; OFT3: percentile 44.5, $z = -0.18$, $P = 5.70 \times 10^{-1}$). **f**, 2D embedding of behavioral flow across all datasets (CSI: $n = 30$, AS: $n = 15$, yohimbine: $n = 15$, CRS: $n = 16$, DREADD: $n = 8$, IFS: $n = 20$, controls (combined): $n = 88$). The crossbars represent the average UMAP1 and UMAP2 values with s.e.m. for each group. **g**, A schematic showing the stratification of nonresponding and responding groups based on BFL score. **h**, log(BFL) scores for control ($n = 15$), nonresponder ($n = 10$) and responder ($n = 10$) animals. **i**, Cluster occurrences in nonresponding ($n = 10$) versus responding ($n = 10$) mice (two-tailed $t$-tests with multiple testing correction). **j**, The absolute differences in behavioral flow in responding versus nonresponding mice. **k**, BFA reveals a group effect between responding and nonresponding mice (one-tailed $z$-test, percentile 99.9, $z = 6.01$, $P = 9.23 \times 10^{-10}$, $d = 3.34$). **l**, Freezing response before (PRE) IFS exposure and during subsequent extinction sessions (Ex1–6) for control ($n = 15$), nonresponder ($n = 10$) and responder ($n = 10$) mice (two-way ANOVA for general effect, followed by two-tailed $t$-tests between groups with multiple testing correction). $P$ values and adjusted $P$ values are denoted as *<0.05, **<0.01 and ***<0.001; n.s., not significant. The error bars in the bar plots denote mean ± s.e.m. For box plots, the center line denotes the median value, while the bounding box delineates the 25th to 75th percentiles. The whiskers represent 1.5 times the interquartile range from the lower and upper bounds of the box.

effects of stress. Distinguishing responders from nonresponders is a great challenge. We thus turned to an experimental setup that allowed us to make predictions about the behavioral response of individual animals. We used an inescapable footshock (IFS) paradigm, which delivers a series of strong footshocks over 20 min and induces long-lasting behavioral changes in rats[23,24] and mice[25,26]. Mice were tested in the OFT the day before stress exposure (OFT1), the day after stress exposure (OFT2) and one week afterward (OFT3). Between OFT2 and OFT3, mice were exposed to one extinction session per day, during which fear memory was assessed (Fig. 5a). Cluster analysis revealed a strong

stress-induced phenotype on OFT2, the day after footshock exposure (Fig. 5b and Supplementary Fig. 1g), and two behavioral transitions revealed a significant difference between groups (two-tailed $t$-tests, adjusted $P < 0.05$; Fig. 5c,d). These effects on cluster occurrences and transitions were not observable before footshock exposure (OFT1; Extended Data Fig. 3a,b) and disappeared again after extinction during OFT3 (Extended Data Fig. 3c,d). These effects were all confirmed with BFA (Fig. 5e). Plotting the behavioral transition dynamics in 2D together with all previous experiments revealed that IFS mice cluster with acute stress and noradrenaline manipulations (Fig. 5f).

Distinguishing responders to stress from nonresponders is often done using a single behavioral measure[27], which raises questions about bias, reliability and validity[28]. To address this, we used the BFL score that, beyond its usefulness for computing effect size (Fig. 1j), can be viewed as a continuous 'treatment-responsivity measure', which contains information about the behavioral change for each individual mouse. Thus, IFS-exposed mice with a BFL score similar to the control group were classified as nonresponders, while animals outside the range of BFL scores of the control group were classified as responders (Fig. 5g,h). Using this new group assignment, we compared OFT2 performance and found—as expected—different cluster representation and behavioral flow (Fig. 5i,j and Supplementary Fig. 1h) and a strong group difference using BFA (one-tailed $z$-test, $P = 9.23 \times 10^{-10}$; Fig. 5k). This group assignment (based on OFT2 performance) allowed a forward prediction, revealing that responders showed a stronger and more protracted freezing response during 6 consecutive days of extinction training (Fig. 5l).

### Transferability across labs and behavioral tests

To expand beyond the OFT, we applied our analysis pipeline to existing video recordings from three different behavioral tests. We also asked if our approach was sufficiently robust to be applied to OFT data acquired from a different laboratory. Due to the change in recording and test setup, we performed a new $k$-means clustering for each experiment, followed by the same analysis pipeline for BFA and BFF as described above.

To apply our pipeline to the marble burying test, mice were either injected (intraperitoneally) with yohimbine (3 mg kg$^{-1}$) or vehicle (Fig. 6a), immediately before exposure to the test cage. We again used 25 clusters as increasing the number of clusters did not increase power (Fig. 6b). Several clusters differed significantly between yohimbine and vehicle injections (two-tailed $t$-tests, adjusted $P < 0.05$; Fig. 6c and Supplementary Fig. 1i), an effect that was confirmed by BFA (Fig. 6d). After visual inspection, we detected a single cluster (cluster M.3; Supplementary Video 15) that represented digging behavior and was strongly reduced in yohimbine animals, in agreement with the total number of buried marbles after the test (Fig. 6e,f). Resolving the digging cluster over 6 min bins, we see a slow onset of the strong drug effect over time (Fig. 6g), consistent with drug uptake dynamics. A 2D embedding (BFF) of binned transition data identified similar temporal dynamics based on the analysis of the complete behavioral flow (Fig. 6h). This method was also able to resolve more subtle behavioral changes within the control animals, which evolve over the 30 min trial and probably capture habituation to the test environment.

Mice were assessed in the light–dark box test 2 days after exposure to CRS (Extended Data Fig. 4a). Power analysis showed that 50 clusters yielded the highest power in detecting group differences (Extended Data Fig. 4b). While no cluster difference survived the multiple testing correction (Extended Data Fig. 4c), BFA did reveal a clear group difference (Extended Data Fig. 4d), and a 2D embedding showed a subtle phenotype (Extended Data Fig. 4e).

To apply our pipeline to a fear conditioning setup, we tracked animals' behavior during the 25 min IFS shock session (Extended Data Fig. 4f). Our analysis pipeline revealed a strong behavioral change in IFS animals using power analysis, cluster usage and BFA (Extended Data Fig. 4g–i). We also investigated behavioral changes over 5 min bins in the fear conditioning box using BFF (Extended Data Fig. 4j). The 2D embedding highlights the strong behavioral change induced in the IFS group after onset of the random shocks (bins 2–5).

Finally, two OFT experiments were conducted in another laboratory (Fig. 6i,j), where the setup differed in many aspects from that used to originally establish the analysis workflow. In the first OFT experiment, mice received either vehicle, or 1, 2 or 3 mg kg$^{-1}$ diazepam per os (Fig. 6i). In the second experiment, mice received either vehicle, or 1, 3 or 6 mg kg$^{-1}$ yohimbine intraperitoneally (Fig. 6j). We performed a new $k$-means clustering using 25 clusters (highest power over all dosages

and experiments; Extended Data Fig. 4l,m) on animals within these two experiments. Classical readouts highlighted clear dose-dependent differences between the two pharmacological compounds (Extended Data Fig. 4k). Cluster usage differed strongly for yohimbine and diazepam injected animals (Fig. 6k,l and Supplementary Fig. 1j). BFA highlighted differences between vehicle and all yohimbine dosage groups (Fig. 6n), whereas for diazepam a dosage-dependent effect was seen only for the two higher dosages (2 and 3 mg kg$^{-1}$; Fig. 6m). The grading of the yohimbine effects was so pronounced that BFA was able to distinguish between all different dosage groups (Extended Data Fig. 4o), whereas diazepam effects could be detected only between 1 mg kg$^{-1}$ versus 2 mg kg$^{-1}$ and 1 mg kg$^{-1}$ versus 3 mg kg$^{-1}$, but not between 2 mg kg$^{-1}$ versus 3 mg kg$^{-1}$ (Extended Data Fig. 4n). Finally, BFF resolved the different dosage-dependent trajectories of the two compounds well (Fig. 6o). Together, these experiments underscore the stability of the method across setups, labs, behavioral test types and phenotypes.

### Discussion

When big data approaches emerge, it often takes years to recognize the problem of false positive findings due to inflated alpha-error probability. This was the case with transcriptomics in biology[29,30], with functional magnetic resonance imaging in neuroscience[31], and with Genome-Wide-Association-Studies in medicine[32]. Currently, big data approaches are revolutionizing behavioral neuroscience[7,8], enabling identification of relevant representations and motifs from behavioral recordings[33–35], and quantification of group differences based on cluster occurrence or transitions[9,11,14,36–39]. However, behavior is notoriously variable[40–43] and the high number of measurements make stringent multiple testing correction quintessential. As a result, large group numbers are required, which runs counter to animal welfare regulations. Our BFA addresses this issue by circumventing the multiple testing problem. Using a large number of independent behavioral datasets, we demonstrate that this approach efficiently assesses treatment effects across various behavioral test setups and is able to resolve subtle phenotypes when classic analyses fail.

A great challenge in biomedical research is to identify treatment responsiveness on the level of individual animals using behavior testing. In stress research, the most popular approaches distinguishing resilient animals (they maintain health) from susceptible animals (they develop disease)[44,45] are based on only a single measure obtained in one single test. For example, social interaction time[27], sucrose consumption[46], the startle response[47] or exploratory behavior during approach–avoidance tests[48,49]. More elaborate strategies use several behavioral tests following stress exposure and dissociate resilient from susceptible mice by assigning composite test scores[50–52]. While the former approach is problematic because it relies on a single measure, the latter is labor intensive and incompatible with some experimental designs. We leverage the ability of BFL to represent high-dimensional behavioral data collected from one behavioral test in a single 'treatment responsiveness score' per animal, which serves as a 'stress-responsivity' measure. This approach enables the integration of behavior analysis with big data approaches, such as molecular screening or high-throughput imaging.

Several notable limitations plague any data-driven attempt to segment behavior recordings. First, not all clusters yield behaviors that can be recognized by human observers. We chose to provide a computationally efficient $k$-means clustering approach, based on features extracted from body point tracking with sensible temporal integration. Many of the resulting clusters yield recognizable and interpretable behavioral motifs, and the observed transitions reveal meaningful behavioral sequences of mouse behavior, but many clusters (and their transitions) remain difficult to interpret. Second, confounding factors and biases can influence each step of the analysis pipeline. Pose estimation is affected by the amount of training data, lighting conditions, occlusion and animal appearance. The subsequent clustering of tracking data depends on the choice of input features and their preprocessing,

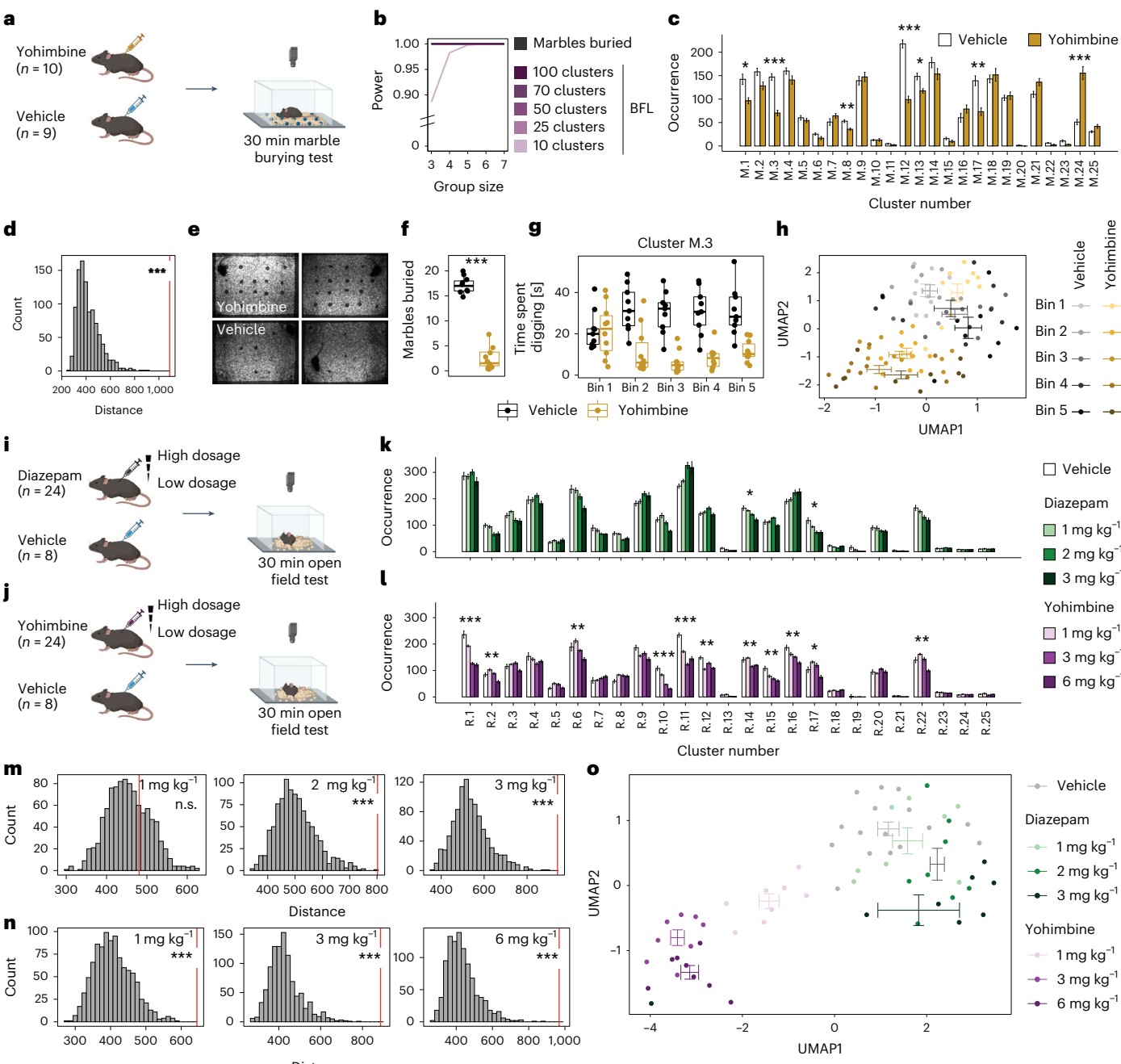

**Fig. 6 | BFA and BFF are transferable to other setups. a**, A schematic showing the experimental design for the marble burying test (MBT) after yohimbine or vehicle injection. **b**, Power analysis comparing different numbers of $k$-means clusters with the number of 'marbles buried' for MBT. **c**, Cluster occurrences in MBT (yohimbine: $n = 10$, vehicle: $n = 9$; two-tailed $t$-tests with multiple testing correction). **d**, BFA reveals a treatment effect of yohimbine (one-tailed $z$-test, percentile 99.9, $z = 7.1$, $P = 6.21 \times 10^{-13}$, $d = 6.57$) in MBT. **e**, Examples of final frames showing marbles buried after yohimbine (top) versus vehicle injection (bottom). **f**, Marbles buried differ significantly (two-tailed $t$-test, $t(16) = 16.2$, $P = 2.03 \times 10^{-11}$) after vehicle ($n = 9$) versus yohimbine ($n = 10$) injection. **g**, Time spent in cluster M.3 (digging cluster) evolves differently over time for mice after vehicle ($n = 9$) or yohimbine ($n = 10$) injection. Each bin represents 6 min. **h**, BFF shows change in behavior profile over time for yohimbine ($n = 10$) and vehicle ($n = 9$). **i**, A schematic showing the experimental design for OFT after diazepam or vehicle administration. **j**, A schematic showing the experimental design for OFT after yohimbine or vehicle injection in another laboratory using a different

OFT setup. **k,l**, Cluster occurrence in diazepam ($n = 24$, vehicle: $n = 8$) (**k**) or in yohimbine ($n = 24$, vehicle: $n = 8$; one-way ANOVA with multiple testing correction) (**l**). **m**, BFA shows treatment effects after higher doses of diazepam (one-tailed $z$-test, 1 mg kg$^{-1}$: percentile 68.93, $z = 0.5$, $P = 3.08 \times 10^{-1}$; 2 mg kg$^{-1}$: percentile 99.9, $z = 4.14$, $P = 1.76 \times 10^{-5}$, $d = 2.72$; 3 mg kg$^{-1}$: percentile 99.9, $z = 4.91$, $P = 4.64 \times 10^{-7}$, $d = 4.2$). **n**, BFA reveals behavioral changes after different doses of yohimbine injections (one-tailed $z$-test, 1 mg kg$^{-1}$: percentile 99.9, $z = 4.44$, $P = 4.52 \times 10^{-6}$, $d = 4.32$; 3 mg kg$^{-1}$: percentile 99.9, $z = 6.1$, $P = 5.27 \times 10^{-10}$, $d = 6.12$; 6 mg kg$^{-1}$: percentile 99.9, $z = 6.37$, $P = 9.60 \times 10^{-11}$, $d = 6.33$). **o**, BFF separates the different doses of diazepam ($n = 24$) and yohimbine ($n = 24$) administrations from vehicle injections (combined: $n = 16$). $P$ values and adjusted $P$ values are denoted as *<0.05, **<0.01 and ***<0.001. The error bars in the bar plots denote mean ± s.e.m. For box plots, the center line denotes the median value, while the bounding box delineates the 25th to 75th percentiles. The whiskers represent 1.5 times the interquartile range from the lower and upper bounds of the box.

the choice of clustering algorithm and hyperparameters. The use of a supervised cluster classifier introduces additional degrees of freedom such as the choice of supervised model and its hyperparameters. The BFA itself may be influenced by the performance of all these previous steps. We do show that our analysis approach detects treatment effects in various behavioral experiments, regardless of clustering algorithm and their corresponding input features, number of clusters and integration period. However, when applying the pipeline to new setups (for example, different arenas, camera angles or frame rates), or when different behaviors emerge, a new clustering is necessary, and the optimal number of clusters needs to be determined. While these considerations challenge the idea of selecting a fixed set of clusters, we also demonstrate that stabilizing clusters can resolve behavioral phenotypes and enable large-scale comparisons of individual animals across experimental conditions. Thus, there is a trade-off between identifying the optimal choice of parameters to boost statistical power for each given experiment and the decision to stabilize clusters to allow direct comparisons across large numbers of datasets collected within a given laboratory or potentially across larger research consortia.

Our work focused on tool development for an in-depth behavioral analysis of single animals; however, we did not measure interactions between multiple animals in this set of experiments. Two unsupervised approaches have recently emerged that provide a versatile platform for analyzing social interactions[12,53]. The tracking data from these tools are compatible with our analyses; thus, our pipeline adds to the rapidly expanding toolbox for nuanced behavioral analyses of single or multiple animals.

## Online content

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

## Methods

### Animals

Mice were maintained in a temperature- and humidity-controlled facility on a 12 h reversed light–dark cycle (lights off at 08:15) with food and water ad libitum. Mice were housed in groups of five per cage and used for experiments when 2.5–4 months old unless stated otherwise. For each experiment, mice of the same age were used in all experimental groups to rule out confounding effects of age. All tests were conducted during the animals' active (dark) phase from 10:00 to 18:00. Mice were habituated to the colony room for at least 2 weeks before experimentation. Mice were single housed 24 h before behavioral testing to standardize their environment and avoid disturbing cagemates during testing[54,55]. All procedures were carried out in accordance to Swiss cantonal regulations for animal experimentation and were approved under licenses ZH155/2015, ZH161/2017, ZH106/2020 and ZH067/2022. The behavior experiments conducted by Roche were carried out under license BS2448.

Detailed information about the behavioral tests and specifics of single experiments can be found in Supplementary Note 3.

### Pose estimation and tracking-based analysis

DeepLabCut 2.0.7 (DLC)[3] was used to track 13 body points of each animal. Tracked points included nose, head center, neck, right ear (earr), left ear (earl), body center, body center left (bcl), body center right (bcr), left hip (hipl), right hip (hipr), tailbase, tail center and tail tip. For the OFT, the marble burying test and the fear conditioning arena, the four corners were tracked additionally to automatically detect the arena boundaries in each recording. For the light–dark box test, six points were tracked corresponding to the corners of the light and dark box. The networks for different tests were trained using 10–20 frames from multiple randomly selected videos for 250,000–1,030,000 iterations. $X$ and $Y$ coordinates of DLC tracking data were imported into R Studio (v3.6.1) and processed with the DLCAnalyzer package[16]. Points relating to the arenas were used to define the arenas in silico by using their median $XY$ coordinates. Values of points with low likelihood (<0.95) and points tracked outside an existence polygon (arena scaled by a factor 1.3 except for the light–dark box test) were removed and interpolated using the R package 'imputeTS' (v3.2). The speed and acceleration of each point was determined by integrating the animal's position over time. The pixel-to-centimeter conversion ratio for each video was determined by comparing the volume of the arena in silico in pixels squared with the measured size of the arena in centimeters squared. Zones of interest were calculated from the arena using polygon-modification functions. Furthermore, we applied a previously trained supervised classifier[16] to quantify supported and unsupported rears in the OFT on a per-frame basis.

### Statistical analysis

To assess group differences based on tracking data before clustering, we used automated metrics from the DLCAnalyzer package[16] such as distance moved, time spent in center, transitions, time in light box and number of supported and unsupported rears. Number of marbles buried was counted manually at the end of each recording. Significance was assessed with a standard parametric $t$-test.

To assess differences in freezing responses for different group assignments (between control, nonresponders and responders), we first performed a two-way repeated measures analysis of variance (ANOVA). If a significant two-way interaction between time point (extinction sessions 1–6) and group assignment emerged, we performed pairwise comparisons using standard parametric $t$-tests. $P$ values were adjusted using the Bonferroni method.

### Generation of feature data for $k$-means clustering

We used the pose estimation data ($N ≈ 15,000$ frames) to generate a set of $m = 41$ features (Supplementary Table 2) resulting in a large numeric matrix $X_i ∈ \mathbb{R}^{N × m}$ for each recording (here denoted with index $i$). Five different types of feature were used: acceleration of points, distance between point pairs, angle between two point pairs, distance of point to closest border, and area of a polygon spanned by multiple points (Supplementary Table 2). Feature data were then normalized on a per-recording level. $z$-Score normalization was used for distances and area features, angle data (in rad) were not normalized, and border proximities and accelerations were scaled linearly (with a factor of 0.1 and 4, respectively). These feature data were furthermore expanded over sequences of ±15 frames ($t = 31$) centered on each frame to generate a larger feature set of $mt = 1,271$ values for each frame resulting in $X_{\text{temporal},i} ∈ \mathbb{R}^{N × mt}$.

### Determining best number of clusters

For determining the best number of clusters, we applied an approach described previously by others[10,14]. We first ran the clustering for a total number of 100 clusters. For each cluster, we then computed the proportion of image frames assigned to it. To determine the best number of clusters, we added the cluster proportions (sorted from high to low proportion) and chose the number of clusters that contain 95% image frames.

### $k$-Means clustering

For $k$-means clustering of OFT data, we selected a random subset of 20 samples per experiment from the CSI, the acute swim stress and the yohimbine injection experiments ($s = 60$). For the fear conditioning box, we selected a random subset of ten samples from the footshock session and each fear extinction day ($s = 70$). All feature data of each frame for these subsets were combined into one single large matrix $X_{\text{clustering}} ∈ \mathbb{R}^{Ns × mt}$ that was then $z$-score normalized across columns. The normalized feature matrix was $k$-means clustered using the function bigkmeans() of the R package 'biganalytics' (v1.1.21). For the light–dark box test and the marble burying test, $k$-means clustering was run on all available samples, respectively. Correlation plots between $k$-means clustering and other clustering algorithms (VAME and B-SOiD) were created using the R package 'corrplots' (v0.92).

### B-SOiD

As a comparison with $k$-means clustering, we ran B-SOiD[9] on the CSI dataset. We followed the steps described on the tutorial webpage (bsoid.org). In short, we trained B-SOiD on a random subsample of 20 files containing the pose estimation computed by DLC. Due to computational issues, we used a reduced set of nine tracking points including nose, head center, neck, body center, bcl, bcr, hipl, hipr and tailbase as input for B-SOiD. We explored different 'minimum cluster size ranges' and ended up with a range between 0.10 and 0.95. The remaining 39 files (not used for clustering) got their clusters assigned using a trained random forest classifier.

### VAME

We ran VAME[14] (v1.1) using PyTorch (v1.7.0) and followed the workflow as described in the publication. For the initial comparison on the CSI experiment, we ran VAME on all 59 pose estimation files from DLC. For the later comparison, we ran VAME on the same subset of 20 samples per experiment as used for the $k$-means clustering. The 11 tracking points including nose, head center, neck, earr, earl, body center, bcl, bcr, hipl, hipr and tailbase used as input were egocentrically aligned to the two tracking points nose and tailbase. Before clustering, we changed the parameters 'n_cluster' to 80 (see above how we determined the best number of clusters) or 25, 'pose confidence' to 0.95 and 'n_features' to 22 in the configuration file.

### Clustering classifier

To transfer clustering to larger or new datasets, we trained a sequential neural network to imitate the clustering results obtained with $k$-means

or VAME. We used the framework designed in a previous publication[16] with the clusters obtained from $k$-means or VAME as ground truth labeling data and $X_{clustering}$ as input data. We used R packages 'reticulate' (v1.24), 'tensorflow' (v2.9.0), and 'keras' (v2.8.0) to design and train the neural network. We used a neural network with a single hidden layer of 1,024 units (using 'relu' activation) and an input shape of 1,271 followed by a dropout layer with a rate of 0.4 to prevent overfitting. We used an output layer with 25 output neurons using the 'softmax' activation function. Data were shuffled before training for 30 epochs with a batch size of 512. We used the 'categorical crossentropy' loss function, 'rmsprop' as optimizer and 'accuracy' as metric during training. Clustering classifiers were then applied to $X_{temporal}$ of each individual recording to obtain the final clustering results.

## Clustering classifier assessment

To assess the performance of the clustering classifiers, we performed a 10-fold cross-validation. All 60 recordings in the clustering (that is, training) set were randomly shuffled before sequentially a different set of six recordings was set aside for validation each time and the training was performed on the remaining 54 recordings only. Then, for each cross-validation pass, we calculated precision, recall and F1 score on a per-cluster basis.

## Label data processing

We used the newly written BehaviorFlow package to process all label data (that is, cluster assigned to each frame) from the $k$-means classifier, VAME and B-SOiD. To remove noise and single frame misclassifications, we processed this data by first smoothing all labels using a sliding window of ±5 frames and selecting the most abundant categorical value across the window using the SmoothLabels_US() function. Next, we calculated metrics such as number of clusters, behavior onset/offsets and time spent in each cluster on a per-recording basis using the CalculateMetrics() function. We then calculated the transition matrix across all label groups for each recording independently using the AddTransitionMatrixData(). This function first removes any repeating labels from each label vector to create an occurrence vector. Then, this occurrence vector and the same vector shifted by one element are used to calculate the contingency table using the table() function of the base package of R resulting in a transition matrix $T_i \in W^{NC \times NC}$ where $i$ denotes the $i$th recording and NC denotes the number of clusters. To calculate stabilized transition matrices $T_{stabilized,i} \in \mathbb{R}^{NC \times NC}$, we used the function CalculateStabilizedTransitions() for defined subsets and control recordings. The stabilized transition matrix is defined as the difference to the mean of the control recordings (equation (1)). Next, we calculated the confusion matrix across all label groups using the AddConfusionMatrix() function. This function calculates a contingency table for a source–target pair using the table() function of the base package of R with the full label vector of the source and the full label vector of the target label group.

$$T_{stabilized,i} = T_i - \sum_C^{N_C} \frac{T_C}{N_C}, \qquad (1)$$

where $T_i$ is the transition matrix of recording $i$, $N_C$ is the number of control group recordings and $T_C$ is the transition matrix of control group recording $C$.

Transitions between clusters are visualized using the R package 'circlize' (v0.4.15).

## Statistical analysis of labeling data (BFA)

We used the TwoGroupAnalysis() function of the BehaviorFlow package for the statistical analysis of label data. This function runs a number of statistical tests to test for group differences on the cluster usage level and the transition level on each label group. For number of cluster occurrences and time spent with clusters, a simple parametric $t$-test followed by a Benjamini–Yekutieli multiple testing correction (using

the function t.test() and p.adjust() of R) was used. The same test was also applied to individual transitions. To test for overall differences across all transitions (referred to as BFA), we first calculated the group-wise mean transition matrix and then the absolute Manhattan distance between the two groups based on these mean matrices (equation (2)). We then used a permutation approach to estimate a null distribution of the intergroup distance from random groupings. We randomly shuffled the group assignment vector 1,000 times and calculated the intergroup distance for each sampling. We then used this to calculate the percentile for nonparametric statistics (equation (3)). Next, we calculated mean and standard deviation from the null distribution and $z$ (equation (4)) for parametric tests. We used the error function to calculate the parametric right-tailed $P$ value from $z$ using the function erf() from the R package 'pracma' (v2.3.8) for the hypothesis true distance > null distribution distances (equation (5)).

$$\text{Manhattan distance} = \sum_{j=1}^{N_{clust}} \sum_{k=1}^{N_{clust}} \left| \sum_C^{N_C} \frac{x_{j,k,C}}{N_C} - \sum_T^{N_T} \frac{x_{j,k,T}}{N_T} \right|, \qquad (2)$$

where $N_{clust}$ is the number of clusters, $N_C$ is the number of control group recordings, $N_T$ is the number of test group recordings and $x_{i,j,r}$ is the number of transitions from cluster $i$ to cluster $j$ in recording $i$.

$$\text{Percentile} = \frac{\sum_{b=1}^{N_{bootstraps}} (\text{distance}_{bootstraps,b} < \text{distance})}{N_{bootstraps} + 1}, \qquad (3)$$

where $N_{bootstraps}$ is the number of bootstraps, $\text{distance}_{bootstraps,b}$ is the Manhattan distance from $b$th bootstrapping and distance is the true group Manhattan distance from equation (2).

$$z = \frac{\text{distance} - \text{mean}(\text{distance}_{bootstraps})}{\text{sd}(\text{distance}_{bootstraps})}, \qquad (4)$$

where distance is the true group Manhattan distance from equation (2), $\text{distance}_{bootstraps}$ is the Manhattan distances obtained from bootstrapping, mean() is the arithmetic mean and sd() is the standard deviation.

$$\text{Right-tailed } P \text{ value} = \frac{1 - \text{erf}(z/\sqrt{2})}{2}, \qquad (5)$$

with $z$ from equation (4), where erf() is the error function.

## BFL

To assess similarity of behavioral flow between samples from two experimental groups (A and B), we used the following method. To calculate the BFL of the $i$th sample, its transition matrix $T_i \in W^{NC \times NC}$ was selected as reference. Next, the transition matrices of the remaining samples were split into two sets based on their true grouping $\{T_{A1}, T_{A2}, \dots T_{AN}\}, \{T_{B1}, T_{B2}, \dots T_{BN}\}$. For each of these sets, the element-wise median transition matrices $M_A$ and $M_B$ are calculated using equation (6)

$$M_{ij} = \text{median}\{T1_{ij}, T2_{ij}, \dots, TN_{ij}\} \qquad (6)$$

for $1 < i < NC$ and $1 < j < NC$.

Next, the Manhattan distance of $T_i$ to $M_A$ and $M_B$ is calculated, resulting in $d_A$ and $d_B$. The BFL score is calculated using equation (7), where a score >0 indicates closeness to group A and a score <0 indicates closeness to group B,

$$\text{BFL score} = \log(d_A / d_B). \qquad (7)$$

## Grouping of responder and nonresponder animals

To group IFS animals into responders and nonresponders, we used the BFL scores (see equation (7)) computed on the transition matrices from the OFT performed 24 h after the IFSs. All animals with a BFL score in

the full range of control BFL scores were classified as nonresponders, while the remainder was classified as responders.

## Effect size calculation and power analysis

To calculate effect size we used Cohen's *d*. For two-group comparisons on classical readouts, individual transitions and individual cluster usage, Cohen's *d* was calculated for each variable independently. To estimate effect size across the entire behavioral flow, the absolute Cohen's *d* was calculated on the basis of the BFL scores (see equation (7)) of individual samples in a two-group comparison.

Power curves from Cohen's *d* values were generated using the 'pwr' R package (v1.3.0). The target significance was set to 0.05, and power values were calculated for a variable *N* (from 3 to 7, 12, 20, 30, 50 or 60).

## In silico sensitivity assay

To create the sensitivity curves, we designed an in silico assay that stepwise reduces group sizes and randomly selects a subset of both groups (ensuring that they are equally sized) before running a two-group comparison. We used group sizes 25, 20, 15, 10 and 5. To better estimate the *P* value at each step, we performed 50 random samplings followed by a two-group comparison each time. Last, we $-\log_{10}$ transformed all *P* values before calculating the mean and standard deviation of the *P* values for each dataset.

## log-linear modeling

To test for an association between yohimbine dosage and the occurrence of transitions between clusters, we used a log-linear model. We first transformed both the yohimbine dosages and the transition occurrences using the natural logarithm. We then fitted a linear model to these two variables using the lm() method from R.

## 2D embedding

For 2D embedding, we used the function Plot2DEmbedding() of the BehaviorFlow Package. As input data, the stabilized transition matrix ($T_{\text{stabilized},i}$) was used for each recording. We then used an UMAP embedding with the function umap() from the R package 'M3C' (v1.16.0). Points were colored by relevant groups, and ± standard error of the mean (s.e.m.) ranges (based on UMAP1 and UMAP2 coordinates) were added to the plot for visual aid.

## Time bin analysis

To split clustering results into time bins (5 × 6 min for marble burying test and 5 × 5 min for fear conditioning), we used the function CreateBinnedData() from the BehaviorFlow Package. The functions CalculateMetrics() and AddTransitionMatrixData() from the same package were then used to compute the cluster usage and transition matrix for each time bin in each recording.

## Reporting summary

Further information on research design is available in the Nature Portfolio Reporting Summary linked to this article.

## Data availability

All video data produced in our lab (707 separate recordings), corresponding pose estimation data and metadata has been deposited online and can be accessed via Zenodo at https://doi.org/10.5281/zenodo.8186065 (ref. 56) and https://doi.org/10.5281/zenodo.11235068 (ref. 57). All video data and pose estimation data produced by Roche (64 separate recordings) can be accessed via Zenodo at https://doi.org/10.5281/zenodo.8188683 (ref. 58) and https://doi.org/10.5281/zenodo.11235915 (ref. 59).

## Code availability

The BehaviorFlow package and any script used to analyze our data and generate the manuscript figures can be accessed freely (GPL-3.0 license) via GitHub at https://github.com/ETHZ-INS/BehaviorFlow.

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

## Acknowledgements

The lab of J.B. is supported by the ETH Zurich, ETH Project Grant ETH-20 19-1, SNSF Grants 310030_172889 and 310030_204372, the ETH 3R-Hub, the Basel Research Centre for Child Health (BRCCH), the Swiss 3R Competence Center, Roche and the Hochschulmedizin Zürich Flagship project STRESS. We thank the staff of the EPIC for the excellent animal care and their service to our animal facility. We thank J. Bode for maintaining the animal colony, R. (Yvonne) Zhang for practical experimental help, S. Leimbacher for contributions to code development and behavior testing, and P.-L. Germain for critical reading of the manuscript and for supporting our data infrastructure. We thank Br. Algeyer for conducting the experiments at Roche and Y.-P. Zhang, D. Roqueiro and F. Tozzi for comments on the manuscript. The scope of Roche funding did not extend to the acute swim stress experiments, and swim stress was employed here as a stressor, not as a measure of despair as conducted in the forced swim test. Figures showing experimental designs or mice icons in general were created with BioRender.com.

## Author contributions

L.M.v.Z.: conceived experiments, developed algorithms, conducted data analyses, produced figures, interpreted results and wrote manuscript. F.K.R.: developed algorithms, conducted data analyses, produced figures, interpreted results and wrote manuscript. O.S.: conducted experiments (CSI, AS, yohimbine and MBTY). R.W.: conducted experiments (CRS (OFT+LDB) and IFS). M.P.: conducted experiment (AS). S.N.D.: conducted experiment (LC-DREADD). E.C.O.: supervised experiments (yohimbine, diazepam) and provided resources. J.B.: conceived experiments, interpreted results, provided resources and funding, and wrote manuscript.

## Funding Information

## Competing interests

E.C.O. was employed by F. Hoffmann-La Roche AG Switzerland at the time of study conduct and manuscript submission. O.S. was funded

by a Roche postdoctoral fellowship. The other authors declare no competing interests.

## Additional information

**Extended data** is available for this paper at https://doi.org/10.1038/s41592-024-02500-6.

**Correspondence and requests for materials** should be addressed to Johannes Bohacek.

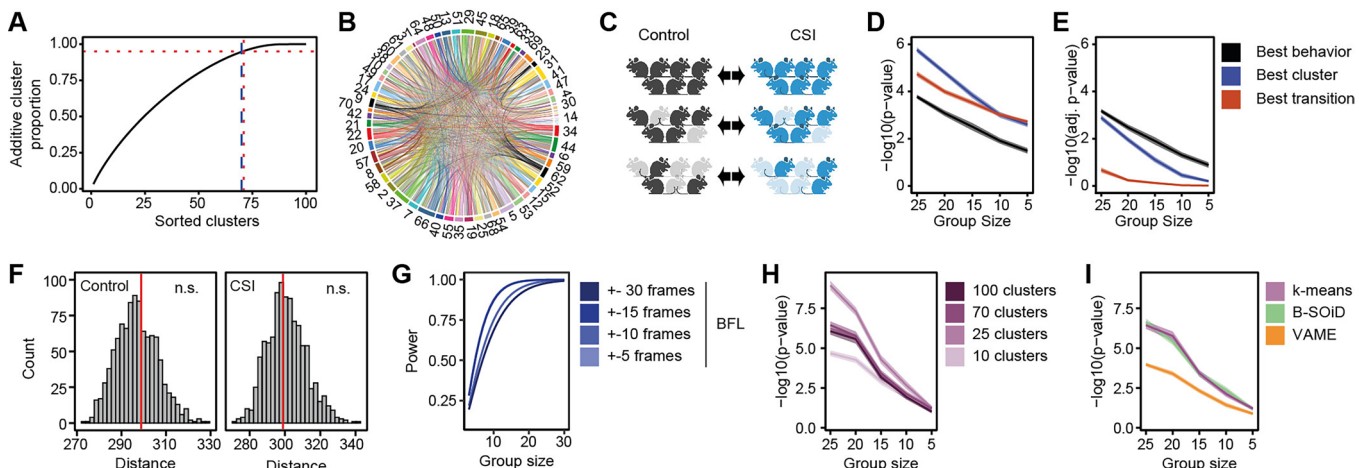

**Extended Data Fig. 1 | BFA increases power to detect phenotypes.**
(**a**) Determining the optimal number of clusters for k-means. Vertical, red-dashed line marks the number of clusters (=71) which represent 95% of all frames (horizontal red-dashed line), and the blue dashed line marks the number of clusters (=70) we used for the CSI analysis. (**b**) Average behavioral flow for all 70 clusters over all animals. (**c**) Schematic of *in silico* approach to generate random subsets of each group of mice to run multiple two-group comparisons while gradually reducing group sizes. (**d**) Phenotype detection sensitivity in CSI with unadjusted p-values (two-tailed *t*-test). Cluster usage and cluster transitions were compared against the best statistical value between distance, time in center, supported rearing and unsupported rearing, termed "best behavior". (**e**) Sensitivity in CSI with adjusted p-values (two-tailed *t*-test) after appropriate multiple testing correction. (**f**) BFA shows no differences for only control (one-tailed *z*-test, percentile=60.3, z=0.2, p=4.19*10$^{-1}$) or only CSI animals (one-tailed *z*-test, percentile=45.0, z=−0.2, p=5.78*10$^{-1}$). (**g**) Power analysis comparing different integration periods. (**h**) BFA (one-tailed *z*-test) enhances sensitivity using various numbers of k-means clusters. (**i**) Phenotype detection sensitivity in CSI using B-SOiD or VAME clustering with BFA (one-tailed *z*-test). The bands in each sensitivity plot display ± SEM.

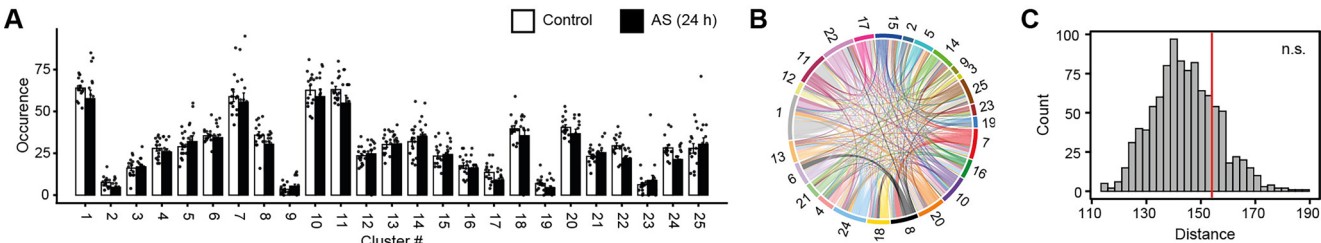

**Extended Data Fig. 2 | Clustering results in AS after 24 hours.** (**a**) Cluster occurrences in AS (24 h: n=15, controls: n=15; two-tailed *t*-tests with multiple testing correction). (**b**) Absolute difference of behavioral flow in control vs. AS (24 h). (**c**) BFA does not show treatment effects at 24 h (one-tailed *z*-test, percentile=80.9, z=0.85, p=0.198). Error bars in the bar plot denote mean ± SEM.

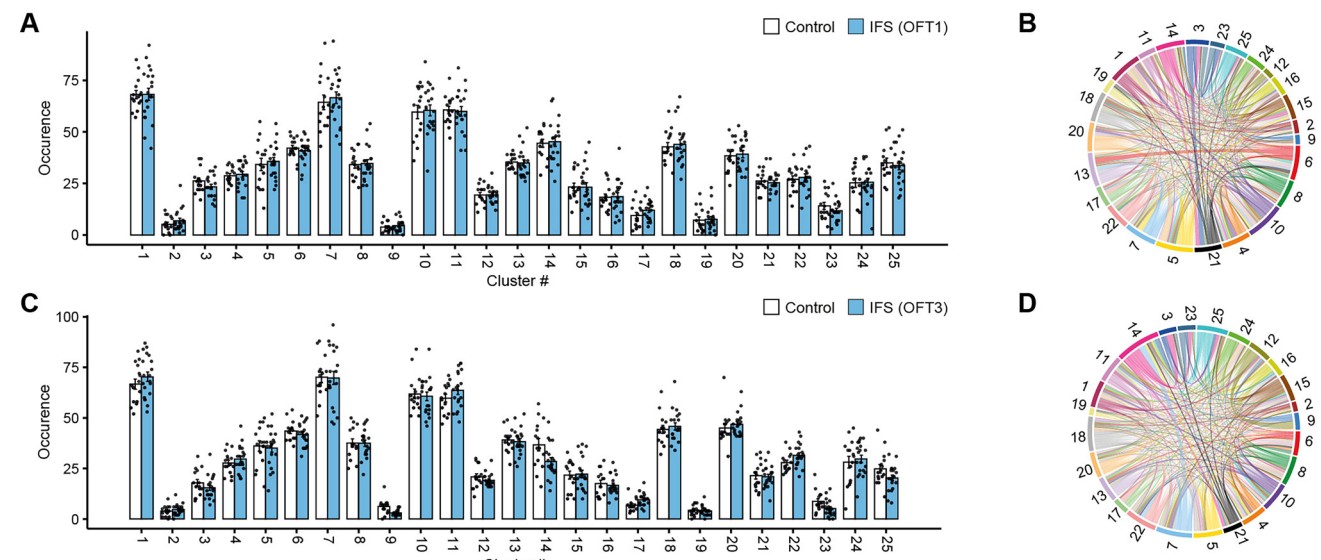

**Extended Data Fig. 3 | Clustering results in IFS (for OFT1 and OFT3). (a)** Cluster occurrences in IFS (OFT1: n=20, controls: n=15; two-tailed *t*-tests with multiple testing correction). **(b)** Absolute difference in behavioral flow in control vs. IFS (OFT1). **(c)** Cluster occurrences in IFS (OFT3: n=20, controls: n=15; two-tailed *t*-tests with multiple testing correction). **(d)** Absolute difference in behavioral flow in control vs. IFS (OFT3). Error bars in the bar plots denote mean ± SEM.

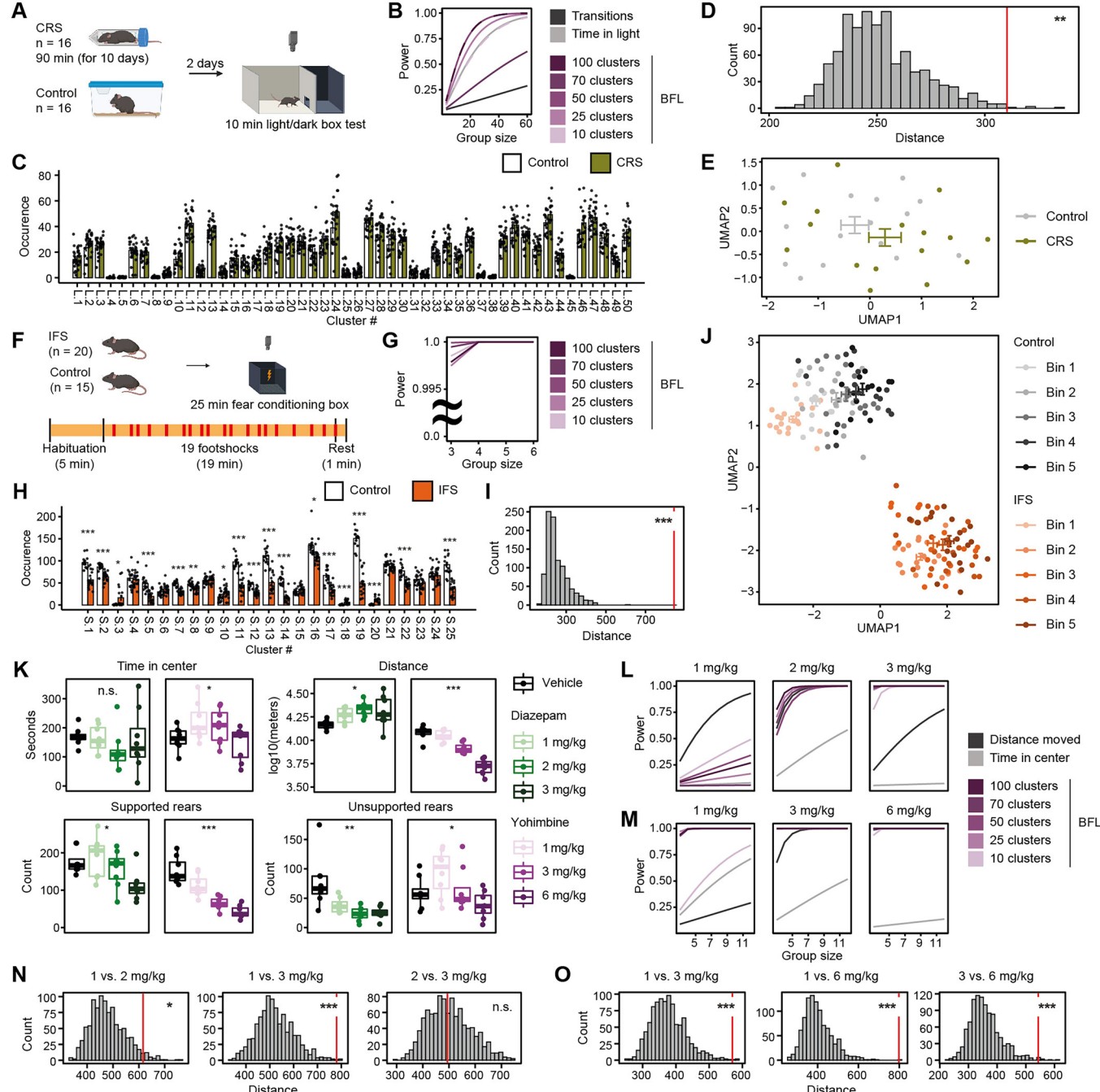

**Extended Data Fig. 4 | See next page for caption.**

**Extended Data Fig. 4 | BFA and BFF applied to other behavioral tests and setups.** (**a**) Schematic showing experimental design for the light-dark box (LDB) test after chronic restraint stress (CRS). (**b**) Power analysis comparing different numbers of k-means clusters with classical readouts ("transitions" and "time in light") for the LDB test. (**c**) Cluster occurrences in LDB after CRS (n=16, controls: n=16; two-tailed $t$-tests with multiple testing correction). (**d**) BFA reveals a treatment effect of CRS (one-tailed $z$-test, percentile=99.5, z=3.06, p=1.09*10$^{-3}$, d=0.91) in LDB. (**e**) BFF applied to LDB data after CRS (n=16, controls: n=16). (**f**) Schematic showing experimental design for exposure to fear conditioning box. (**g**) Power analysis comparing different numbers of k-means clusters for fear conditioning box. (**h**) Cluster occurrences in fear conditioning box (IFS: n=20, controls: n=15; two-tailed $t$-tests with multiple testing correction). (**i**) BFA shows a treatment effect of the fear conditioning box (one-tailed $z$-test, percentile=99.9, z=10.6, p=0, d=5.44). (**j**) BFF applied to fear conditioning box data (IFS: n=20, controls: n=15). Each bin represents 5 minutes. (**k**) Comparison of classical behavior readouts for different doses of diazepam (n=24, vehicle: n=8) or yohimbine (n=24, vehicle: n=8) (one-way ANOVA, time in center for diazepam: F(3,28)=0.78, adj. p=5.14*10$^{-1}$; for yohimbine: F(3,28)=3.2, adj. p=3.84*10$^{-2}$; distance moved for diazepam: F(3,28)=4.09, adj. p=2.62*10$^{-2}$; for

yohimbine; F(3,28)=46.27, adj. p=2.81*10$^{-10}$; supported rears for diazepam: F(3,28)=4.68, adj. p=2.26*10$^{-2}$, for yohimbine: F(3,28)=29.18, adj. p=1.53*10$^{-8}$; unsupported rears for diazepam: F(3,28)=8.85, adj. p=1.38*10$^{-3}$; for yohimbine: F(3,28)=4.02, adj. p=2.11*10$^{-2}$). (**l**) Power analysis comparing different numbers of k-means clusters with classical OFT readouts after treatment with diazepam or (**m**) yohimbine. (**n**) BFA reveals differences between higher doses of diazepam (2 or 3 mg/kg) compared to lower doses (1 mg/kg) (one-tailed $z$-test, 1 vs. 2 mg/kg: percentile=96.4, z=2.04, p=2.06*10$^{-2}$, d=0.19; 1 vs. 3 mg/kg: percentile=99.7, z=3.34, p=4.15*10$^{-4}$, d=1.15; 2 vs. 3 mg/kg: percentile=49.45, z=−0.1, p=5.41*10$^{-1}$). (**o**) BFA shows treatment differences between different doses of yohimbine (one-tailed $z$-test, 1 vs. 3 mg/kg: percentile=99.7, z=3.71, p=1.05*10$^{-4}$, d=2.45; 1 vs. 6 mg/kg: percentile=99.8, z=5.68, p=6.82*10$^{-9}$, d=5.36; 3 vs. 6 mg/kg: percentile=99.2, z=3.43, p=3.00*10$^{-4}$, d=1.49). p-values and adj. p-values are denoted as: *<0.05, **<0.01, ***<0.001. Error bars in the bar plots denote mean ± SEM. For every UMAP embedding, the crossbars represent the average UMAP1 and UMAP2 values with SEM for each group. For box plots, the center line denotes the median value, while the bounding box delineates the 25th to 75th percentiles. Whiskers represent 1.5 times the interquartile range from the lower and upper bounds of the box.

# Reporting Summary

## Statistics

For all statistical analyses, confirm that the following items are present in the figure legend, table legend, main text, or Methods section.

| n/a | Confirmed | |
|-----|-----------|---|
| ☐ | ☒ | The exact sample size (*n*) for each experimental group/condition, given as a discrete number and unit of measurement |
| ☐ | ☒ | A statement on whether measurements were taken from distinct samples or whether the same sample was measured repeatedly |
| ☐ | ☒ | The statistical test(s) used AND whether they are one- or two-sided<br>*Only common tests should be described solely by name; describe more complex techniques in the Methods section.* |
| ☐ | ☒ | A description of all covariates tested |
| ☐ | ☒ | A description of any assumptions or corrections, such as tests of normality and adjustment for multiple comparisons |
| ☐ | ☒ | A full description of the statistical parameters including central tendency (e.g. means) or other basic estimates (e.g. regression coefficient) AND variation (e.g. standard deviation) or associated estimates of uncertainty (e.g. confidence intervals) |
| ☐ | ☒ | For null hypothesis testing, the test statistic (e.g. *F*, *t*, *r*) with confidence intervals, effect sizes, degrees of freedom and *P* value noted<br>*Give P values as exact values whenever suitable.* |
| ☒ | ☐ | For Bayesian analysis, information on the choice of priors and Markov chain Monte Carlo settings |
| ☒ | ☐ | For hierarchical and complex designs, identification of the appropriate level for tests and full reporting of outcomes |
| ☐ | ☒ | Estimates of effect sizes (e.g. Cohen's *d*, Pearson's *r*), indicating how they were calculated |

*Our web collection on statistics for biologists contains articles on many of the points above.*

## Software and code

Policy information about availability of computer code

| Data collection | Video recording:<br>TSE multi conditioning system (TSE Systems Ltd, Germany) |
|---|---|
| Data analysis | Pose estimation:<br>Deeplabcut (v2.0.7)<br><br>Unsupervised Clustering:<br>VAME (v1.1) using PyTorch (v1.7.0)<br>B-SOiD (v2.0)<br><br>Analysis:<br>R (v3.6.1)<br>DLCAnalyzer (v1.0)<br>BehaviorFlow (v1.0)<br><br>R packages and version numbers used for analysis:<br>"pwr" v1.3.0<br>"circlize" v0.4.15<br>"imputeTS" v3.2<br>"M3C" v1.16.0<br>"pracma" v2.3.8 |

```
"corrplot" v0.92
"keras" v2.8.0
"tensorflow" v2.9.0
"biganalytics" v1.1.21
"reticluate" v1.24

All code for this project has been deposited on github and is freely accessible under https://github.com/ETHZ-INS/BehaviorFlow.
```

For manuscripts utilizing custom algorithms or software that are central to the research but not yet described in published literature, software must be made available to editors and reviewers. We strongly encourage code deposition in a community repository (e.g. GitHub). See the Nature Portfolio guidelines for submitting code & software for further information.

## Data

Policy information about availability of data

All manuscripts must include a data availability statement. This statement should provide the following information, where applicable:
- Accession codes, unique identifiers, or web links for publicly available datasets
- A description of any restrictions on data availability
- For clinical datasets or third party data, please ensure that the statement adheres to our policy

All video data produced in our lab (707 separate recordings), corresponding pose estimation data and metadata has been deposited online and can be accessed under https://zenodo.org/record/8186065 and https://zenodo.org/records/11235068. All video data and pose estimation data produced by Roche (64 separate recordings) can be accessed under https://zenodo.org/record/8188683 and https://zenodo.org/records/11235915.

## Human research participants

Policy information about studies involving human research participants and Sex and Gender in Research.

| Reporting on sex and gender | N/A |
| --- | --- |
| Population characteristics | N/A |
| Recruitment | N/A |
| Ethics oversight | N/A |

Note that full information on the approval of the study protocol must also be provided in the manuscript.

# Field-specific reporting

Please select the one below that is the best fit for your research. If you are not sure, read the appropriate sections before making your selection.

☒ Life sciences          ☐ Behavioural & social sciences          ☐ Ecological, evolutionary & environmental sciences

For a reference copy of the document with all sections, see nature.com/documents/nr-reporting-summary-flat.pdf

# Life sciences study design

All studies must disclose on these points even when the disclosure is negative.

| Sample size | Sample size required to detect phenotypes with different analysis methods was determined with an in silico sub-sampling scheme as described in the manuscript. However, datasets were obtained prior to any of these calculations and sample sizes for these datasets were independently determined using power analyses in a number of other projects of the lab. |
| --- | --- |
| Data exclusions | No data were excluded for this study. |
| Replication | For reproducibility assessment we used a number of independently performed experiments that induce acute stress through different means. Additionally, we conducted two independent experiments (diazepam or yohimbine injections) in an external lab (Roche Pharma) to demonstrate transferability of our methods. All attempts of replication were successful. |
| Randomization | For all experiments we used a block design, where animals from all groups were evenly distributed into multiple smaller blocks, each containing one biological replicate of each group. Processing order was randomized within blocks. During all experimental manipulations the block design was strictly followed to prevent introduction of any uncontrollable, experiment-independent effects. |
| Blinding | Investigators were blinded during data collection thanks to the randomized block design. Furthermore, the analysis pipeline algorithms (pose estimation, clustering, clustering classifier training, clustering classifier application and calculation of all other metrics such as number of cluster onset/offset/time and transition matrices) are completely blind to group allocation up to the time of statistical testing and annotation on 2D embeddings. |

# Reporting for specific materials, systems and methods

We require information from authors about some types of materials, experimental systems and methods used in many studies. Here, indicate whether each material, system or method listed is relevant to your study. If you are not sure if a list item applies to your research, read the appropriate section before selecting a response.

## Materials & experimental systems

| n/a | Involved in the study |
|-----|----------------------|
| ☒ | ☐ Antibodies |
| ☒ | ☐ Eukaryotic cell lines |
| ☒ | ☐ Palaeontology and archaeology |
| ☐ | ☒ Animals and other organisms |
| ☒ | ☐ Clinical data |
| ☒ | ☐ Dual use research of concern |

## Methods

| n/a | Involved in the study |
|-----|----------------------|
| ☒ | ☐ ChIP-seq |
| ☒ | ☐ Flow cytometry |
| ☒ | ☐ MRI-based neuroimaging |

## Animals and other research organisms

Policy information about studies involving animals; ARRIVE guidelines recommended for reporting animal research, and Sex and Gender in Research

| | |
|---|---|
| Laboratory animals | C57BL/6J (C57BL/6JRj) mice obtained by Janvier (France) were used for most experiments. Heterozygous C57BL/6-Tg(Dbh-icre)1Gs mice were used for chemogenetic experiments and C57BL/6J mice obtained from Charles River Laboratories (Saint Germain sur l'Arbresle, France) were used for experiments performed at Roche Pharma. Mice from all strains were used for experiments when 2.5–4 months old. |
| Wild animals | The study did not involve wild animals. |
| Reporting on sex | A number of experiments were performed with male mice exclusively, one experiment with female mice exclusively and one experiment was balanced with the same number of male and female mice. Metadata in online repositories contain a complete record on sex for each biological replicate. 2D embeddings revealed no apparent sex based effects in experiments performed with both male and female mice. A lot of the experiments contained in this study were performed over several years in our lab, and the lack of sex based designs is due to requirements of other projects in the lab. |
| Field-collected samples | The study did not involve field-collected samples. |
| Ethics oversight | All procedures were carried out in accordance to Swiss cantonal regulations for animal experimentation and were approved under licenses: ZH155/2015, ZH161/2017, ZH106/2020, ZH067/2022. Ethical approval for behavior experiments conducted by Roche was provided by the Federal Food Safety and Veterinary Office of Switzerland. All animal experiments were conducted in strict adherence to the Swiss federal ordinance on animal protection and welfare as well as according to the rules of the Association for Assessment and Accreditation of Laboratory Animal Care International (AAALAC), and with the explicit approval of the local veterinary authorities (License BS2448). |

Note that full information on the approval of the study protocol must also be provided in the manuscript.

