## [Peer Review File · Nature Methods]

Analysis of behavioral flow resolves latent phenotypes

Corresponding Author: Professor Johannes Bohacek

A version of this paper was originally rejected for publication by Nature Methods, however that decision was reconsidered after appeal by the authors.

Version 0:

Decision Letter:

8th Dec 2023

Dear Professor Bohacek,

Let me first sincerely apologize for the delays in the review process. I understand how frustrating this has been for you. Your Article entitled "Analysis of behavioral flow resolves latent phenotypes" has now been seen by three reviewers, whose comments are attached. While they find your work of potential interest, they have raised serious concerns which in our view are sufficiently important that they preclude publication of the work in Nature Methods, at least in its present form.

As you will see, while the reviewers appreciate that your method addresses an important problem in the field, the reviewers nevertheless raise technical concerns, concerns about unsupported claims and one of the reviewers even requests extension of the methodology to be able to detect temporal changes in behavior or to study social situations.

Should further experimental data allow you to address these criticisms we would be willing to look at a revised manuscript (unless, of course, something similar has by then been accepted at Nature Methods or appeared elsewhere). This includes submission or publication of a portion of this work somewhere else. We hope you understand that until we have read the revised paper in its entirety we cannot promise that it will be sent back for peer-review.

Although we cannot publish your paper, it may be appropriate for another journal in the Nature Portfolio. If you wish to explore the journals and transfer your manuscript please use our manuscript transfer portal. You will not have to re-supply manuscript metadata and files, unless you wish to make modifications. For more information, please see our [manuscript transfer FAQ](http://www.nature.com/authors/author_resources/transfer_manuscripts.html?WT.mc_id=EMI_NPG_1511_AUTHORTRANSF&WT.ec_id=AUTHOR) page.

If you are interested in revising this manuscript for submission to Nature Methods in the future, please contact me to discuss your appeal before making any revisions. Specifically, please do send me a revision plan with your appeal. Otherwise, we hope that you find the reviewers' comments helpful when preparing your paper for submission elsewhere.

I would also like to renew my apologies about the unusually long review process.

Best regards,
Nina

Nina Vogt, PhD
Senior Editor
Nature Methods

Reviewers' Comments:

Reviewer #1:
Remarks to the Author:

In the current manuscript the authors tackle an important problem in the field of behavioral biomedical research of rodents, that is the comparison of complex behaviors and rich behavioral profiles across experimental groups and experiments. They introduce two dimensionality reduction pipeline termed Behavioral Flow Analysis (BFA) and subsequent Behavioral Flow Fingerprinting (BFF) for the assessment of group effects (BFA) and individual variability (BFF) in behavioral data sets obtained in open field test settings. Applying these approaches to different experiments and data sets, they demonstrate that these dimensionality reduction approaches are capable of detecting group and individual differences with higher sensitivity compared to previous methods. I believe that this will be an important methodological addition for the behavioral neuroscience field. I have a number of comments and questions, that I would like the authors to address or clarify.

a. A major drawback of the current approach is the fact that all temporal information of the behavioral dynamics of the animals is lost. Open Field test situations are short and treated as individual units. This will be hampering application of these analysis pipelines, as specifically the change of behavior over time is a crucial feature in many experimental settings and research questions. Can the authors still implement this and test if BFA and BFF are capable to detect behavioral changes over time?

b. A second limitation is in my view the limitation to individual animals in the sparsest of environments (empty open field), which drastically limits the behavioral repertoire of the animals. Implementing multi-animal testing, which will include the animals' social behavior, as well as testing in more complex environments, would significantly improve the usability of these analysis pipelines.

c. When comparing the effectiveness of BFA to other methods, the authors rely on the p-values (e.g., Fig 1J, K, L), which in my view is not entirely correct. While a p-value is informative whether or not an effect exists, it will not reveal the size of the effect. The authors should therefore rather use effect sizes for these comparisons.

d. In the BFF analyses (e.g., Fig. 3 G, H, I) there is still a large overlap between the groups and the directionality of change is not unidirectional. In 3H, for example, there seem to be subgroups of AS (45 min) mice that cluster together and are quite distinct. Do these subgroups of animals also show distinct behavioral profiles, potentially indicating different coping strategies in response to the stress exposure?

e. When comparing flow diagrams (e.g. Figure 4D) the authors claim that these appear highly similar. Can they provide a quantification of this similarity? Making sense of these flow diagrams by visual inspection is currently difficult, due to the multiple transitions between clusters. It might be helpful to implement a threshold and omit cluster transitions that occur with a low likelihood.

f. When testing how well the analysis pipeline can be transferred to new datasets, the authors pick two clusters (1 and 11), which seem to describe similar behaviors. However, it is in my view not sufficient and convincing to only pick 2 out of 25 clusters. For the authors to conclude reproducible clustering transfer for completely new datasets, this needs to be shown and quantified for all 25 clusters.

Reviewer #2:

Remarks to the Author:

"Analysis of behavioral flow resolves latent phenotypes" by von Ziegler et al. presents a method of identifying behavioral clusters and analysing their transitions. The core idea of the paper is that by comparing the distance between a vector of transition probabilities rather than the probabilities themselves individually, the power to detect group differences in behavior is increased.

The authors make several claims about the advantages of their method, but these are not well supported. The broader claims about how this method fits with previous behavior analysis methods is also not sufficiently justified and in some places contradictory.

Main claims

1) Multiple testing in high-dimensional behavior analysis reduces power.

The authors show that this is true on their data using their representation of behavior and a 'classic' representation of open field data. This supports the step of their analysis where they combine transition probabilities into a single distance measure. However, success at this step doesn't support their representation itself. For example, you could apply this step to any behavior representation that includes transitions between behaviors (which is most representations of behavior). It should therefore be made clearer that the potential advantages of this step do not necessarily depend on the rest of the analysis and could equally be applied to, say, MoSeq.

This is really the central claim of the paper (the other two claims are not justified). However, to demonstrate its usefulness, it would be best to compare it to other ways of comparing transition matrices such as divergence statistics.

2) Transferring clusters between experiments can be a challenge that is solved using a 'classifier-in-the-middle' approach.

As I understand it, the authors train a neural network classifier to predict cluster membership using k-means clusters as the ground truth. The output from k-means is a point in space that defines a cluster. Points are assigned to a cluster based on which cluster 'center' they are closest to. The ground-truth best cluster to assign a point to is therefore the closest one. Because the closest cluster center to a new point can just be calculated directly, this direct approach will always be more accurate (it is already perfect according to the setup of the problem), I don't see what the neural network adds here. In places where the neural network disagrees with the simple procedure, it is basically by definition because the neural network has made a mistake.

Furthermore, this also shows that the premise of claim one is not really correct. For a given set of clusters, a new point from a different experiment can be assigned quickly and easily. There is an issue with clusters changing fundamentally from one experiment to the next, but the 'classifier-in-the-middle' doesn't change that. As the authors do themselves in the last section, one way to address this problem is just to redo the clustering on the new data. Again, the classifier is unnecessary.

In case I've completely missed the point, the authors should at least consider making the explanation of the training and ground truth clearer.

3) The problem of individual differences

Individual differences are indeed prevalent in behavior data. However, the proposed solution of considering individual behavioral fingerprints in a high-dimensional space is not unique to the current method. Placing individuals in a high-dimensional space for comparison can be applied to any behavioral fingerprinting approach (quantitative fingerprinting from tracking data dates back at least to 2002 <https://pubmed.ncbi.nlm.nih.gov/12191753/>).

Furthermore, the basic claim that their method improves statistical power relies on ignoring these details (at the expense of interpretability). It is a bit of a contradiction to claim the same method improves power by combining transitions into a single measure and also improves interpretability by not combining transitions into a single measure.

In terms of how the fingerprinting approach is implemented here, it is not recommended to do statistics on UMAP dimensions. Reducing the dimensionality down to 2 and then quantifying the mean and standard error can lead to difficulties in interpretation because different projections can give very different results. Repeating the UMAP with different seeds is insufficient to address this issue. See <https://journals.plos.org/ploscompbiol/article?id=10.1371/journal.pcbi.1011288> for some more extreme examples.

Specific points

- "The majority of preclinical biomedical research is conducted in mice". Probably more accurate to say that the majority of *in vivo* preclinical biomedical research is done in mice.

- intro, first paragraph: "the advent of pose estimation" isn't necessarily the right transition to emphasize. For example, even with centroid tracking, high-dimensional fingerprinting of mouse behavior can be done. Kafkafi et al. have even shown that it can sensitively phenotype drug treated animals in open field assays. See <https://www.ncbi.nlm.nih.gov/pmc/articles/PMC8056474/> as well as their two earlier papers introducing the method in 2008 and 2009. These papers should be cited and discussed in the context of the new method.

- "However, this would entail re-running the unsupervised clustering every time new data is added and soon become computationally challenging as previously observed". There are approaches to k-means that scale better with large datasets. In any case, as discussed above, the existing clusters can be kept and new data assigned to the right cluster with little computation even for large datasets.

- "As a result, large group numbers are required, which runs counter to the goal that these approaches set out to achieve in the first place." Some methods of detailed phenotyping may demonstrate increased statistical power (and therefore justify a reduction in animals), but I wouldn't say reducing animal numbers is 'the', or even the main, goal of behavioral fingerprinting.

Reviewer #3:

Remarks to the Author:

The paper aims to quantify behavioral differences between groups of mice under different conditions. The authors propose a pipeline that takes in behavior clusters (from existing approaches such as VAME, BSOID, etc.), then computes Manhattan distance between cluster transitions of the group means, and perform bootstrapping to estimate null distribution of group distances. The authors then use the right-tailed z-test to compare the actual distance to the null distribution to detect group differences. The authors name this pipeline behavioral flow analysis (BFA). Experiments are performed on a set of mouse datasets (CSI, acute swim, yohimbine) + one more from a different lab, to show that BFA is able to identify group differences across settings. The authors also propose to use this pipeline to measure individual animal differences as well (behavioral flow fingerprinting BFF). I think the paper tackles an important problem, but I have some questions about the approach and experiments:

The paper claims that the "classifier-in-the-middle" approach improves transferability in the introduction. However, the classifier still needs to be trained on clusters in each experimental setting in a supervised way. The authors note in the introduction that "it is difficult to transfer a model between different testing conditions, datasets or laboratories" and claims that the supervised classifier addresses this; however, since the supervised model also needs to be trained in each setting (similar to the clustering approach), it is unclear to me if the claim is true.

Another question I have from the paper is the evaluation setup - it seems BFA almost always results in showing treatment effects (the only test with no effect is one experiment in Fig 5E). The number of negative tests is thus very small (let me know if I might have missed others in the paper). Have the authors done more checks on the method, for example, by taking two similar groups (both with/without treatment) to show that BFA also finds this to be the case?

Additionally, there lacks comparison of the proposed approach to baseline methods, so it is difficult to evaluate the significance of this pipeline. There is one comparison in Supplementary Figure 1D compared to using clustering or transition matrices alone; but the experiment is not comprehensive, since it is (1) only on one of the subsets of the data, (2) unclear if the clustering/transition matrices would perform better with different numbers of clusters or hyperparameters. Another factor that makes comparing the experiments difficult is that the authors have used different input clustering approaches in different data subsets as input to BFA - is there any comparison of which clustering approach should be used with BFA?

While there are discussions on other works in clustering, other methods with similar goals/settings are not discussed (for example, contrastive analysis [A], representation learning ([B,C])). These methods also output representations that could be clustered. I don't think it's necessary to run experiments on everything as baselines, but more discussions of other works in these areas would be helpful to situate this method with respect to related works.

Finally, this method relies on input clusters (which itself relies on estimated keypoints...). This is potentially a brittle pipeline, as it relies on many previous computation steps, some of which are hard to evaluate (for example, it is difficult to determine the "correct" number of clusters and clustering hyperparameters). Additional experiments, such as ablation studies on the cluster quality and cluster number / other hyperparameter variations, would help demonstrate the ability of this pipeline to be applied across different lab settings.

[A] Weinberger et al., Feature Selection in the Contrastive Analysis Setting

[B] Sun et al., Task Programming: Learning Data Efficient Behavior Representations

[C] Azabou et al., Learning Behavior Representations Through Multi-Timescale Bootstrapping

** For Nature Portfolio general information and news for authors, see <http://npg.nature.com/authors>.

Version 1:

Decision Letter:

19th Jan 2024

Dear Professor Bohacek,

Thank you for your letter asking us to reconsider our decision on your Article, "Analysis of behavioral flow resolves latent phenotypes". After careful consideration we have decided that we are willing to consider a revised version of your manuscript. The manuscript should be revised as outlined in your appeal. However, we don't think it is necessary to show that your approach can be extended to social situations (or robotic interactions).

- * include a point-by-point response to our referees and to any editorial suggestions
- * please underline/highlight any additions to the text or areas with other significant changes to facilitate review of the revised manuscript
- * address the points listed described below to conform to our open science requirements
- * ensure it complies with our general format requirements as set out in our guide to authors at www.nature.com/naturemethods
- * resubmit all the necessary files electronically by using the link below to access your home page

Link Redacted

We hope to receive your revised paper within 2-3 months. If you cannot send it within this time, please let us know. In this event, we will still be happy to reconsider your paper at a later date so long as nothing similar has been accepted for publication at Nature Methods or published elsewhere.

OPEN SCIENCE REQUIREMENTS

REPORTING SUMMARY AND EDITORIAL POLICY CHECKLISTS

When revising your manuscript, please submit reporting summary and editorial policy checklists.

DATA AVAILABILITY

CODE AVAILABILITY

Please include a "Code Availability" subsection in the Online Methods which details how your custom code is made available. Only in rare cases (where code is not central to the main conclusions of the paper) is the statement "available upon request" allowed (and reasons should be specified).

MATERIALS AVAILABILITY

ORCID

Nature Methods is committed to improving transparency in authorship. As part of our efforts in this direction, we are now requesting that all authors identified as 'corresponding author' on published papers create and link their Open Researcher and Contributor Identifier (ORCID) with their account on the Manuscript Tracking System (MTS), prior to acceptance. This applies to primary research papers only. ORCID helps the scientific community achieve unambiguous attribution of all scholarly contributions. You can create and link your ORCID from the home page of the MTS by clicking on 'Modify my Springer Nature account'. For more information please visit <http://www.springernature.com/orcid>.

Best regards,
Nina

Nina Vogt, PhD
Senior Editor
Nature Methods

Version 2:

Decision Letter:

Our ref: NMETH-A53539B

27th Jun 2024

Dear Johannes,

Thank you for submitting your revised manuscript "Analysis of behavioral flow resolves latent phenotypes" (NMETH-A53539B). It has now been seen by the original referees and their comments are below. The reviewers find that the paper has improved in revision, and therefore we'll be happy in principle to publish it in Nature Methods, pending minor revisions to satisfy the referees' final requests and to comply with our editorial and formatting guidelines.

Please make sure that the remaining concerns of the reviewers are addressed in full and provide a rebuttal. I also recommend having a colleagues outside of the field read the manuscript and provide some feedback to ensure that the manuscript is accessible to a broad audience.

TRANSPARENT PEER REVIEW

ORCID

Best regards,
Nina

Nina Vogt, PhD
Senior Editor
Nature Methods

Reviewer #1 (Remarks to the Author):

The authors thoughtfully addressed all of my previous comments and incorporated significant improvements in the revised

version of the manuscript. With these changes and additions the manuscript has in my view significantly improved and I can now recommend publication. I believe that the presented analysis pipeline will be of high interest in the behavioral neuroscience community.

Reviewer #2 (Remarks to the Author):

The authors have addressed my specific concerns. I'm not aware of any remaining technical issue in the paper.

I realise the paper represents a lot of work and it seems like a good analysis pipeline, but it exists in the context of many other good analysis pipelines for behavior data. I don't see the kind of methodological advance I would expect for a paper in Nature Methods.

Specifically:

- 1) Clustering to identify states in behavioral data is now common and the authors use several published methods.
- 2) The idea of comparing transition matrices is also not new even if the specific bootstrapping approach is. For example, this article uses transition matrices to compare behaviors <https://doi.org/10.3758/BF03192788> and cites a tool written to compare behavioral transition matrices (https://brill.com/view/journals/beh/125/3-4/article-p157_1.xml) from the 90s.
- 3) I understand why the authors use the classifier, but it still seems unnecessary, or at least a solution to a problem that could be solved another way more simply. As the authors say in their response, they could save the normalization metrics and share those to enable new data to be embedded. They then say that some clustering methods would require the full dataset to re-embed new data, but if comparing across experiments is a priority, the authors could solve this by sticking with k-means.

Reviewer #3 (Remarks to the Author):

I'd like to thank the authors for the detailed feedback to reviewers and updated manuscript with additional experiments. The paper is definitely stronger compared to the initial submission. The modifications include the behavioral flow likeness score, new discussions (ex: different clustering approaches), and additional experiments suggested by reviewers (ex: showing whether the method detects differences when there's none). I'm focusing my review on the new additions as well as comments/responses from other reviewers:

- Classifier-in-the-middle: I appreciate the clarifications from the authors on the classifier-in-the-middle approach, and I see now it is a supervised classifier trained on the cluster data (potentially from many sources). Since the classifier is trained in a supervised way, this approach inherits the limitations of supervised learning (ex: limited ability to generalize to new data, overconfidence on out-of-distribution data). To address this, the authors could assess the classifier's performance on different train/test settings (e.g., using different experiments as train/test), a wider range of datasets, and/or better define limitations and what constitutes a "new setup" requiring classifier retraining.
- I read the author response on the comment about pipeline brittleness and I have remaining concerns on this point. The pose estimator, clustering algorithm, and BFA (or variants) each add more hyperparameters and steps to the analysis. For instance, the performance of the pose estimator can be affected by factors such as occlusion, lighting conditions, and individual differences in animal appearance. The clustering algorithm's performance depends on the choice of hyperparameters and the quality of the input features (as well as choice of pre-processing steps). Additionally, the BFA method itself relies on the accuracy and stability of the clusters identified in the previous steps. While the authors demonstrate their pipeline in different rodent experiments & with a few clustering algorithms, it's important to acknowledge that the framework might not generalize well to situations where tracking is unreliable or when the behavioral repertoire differs significantly from the training data (ex: if classifier-in-the-middle is used). Finally, simply because a method is widely used, doesn't mean it is reliable (especially under all distributions of data/behaviors). To address these concerns, I suggest the authors provide a more thorough discussion of the potential limitations and failure modes of their pipeline.
- UMAP: Another reviewer raised a great point concerning UMAP (and its limitations). I took a closer look and agree with the points raised by the other reviewer. Additionally, UMAP itself has a lot of parameters and the embeddings are highly sensitive to these choices (ex: <https://umap-learn.readthedocs.io/en/latest/parameters.html>). The authors mentioned in the response they no longer use UMAP, but I still see a lot references of BFF with UMAP in the manuscript. Not sure if I've mis-understood the author response on this, or whether the paper has not been updated yet.
- In the newest revision, the paper introduces several new terms (BFA, BFF, BFL) that could be confusing for readers and do not clearly convey the underlying methodology. More descriptive names would improve clarity and understanding. For example: BFA = "Manhattan distance-based bootstrapping" or a similar phrase that highlights the statistical methods used. BFF = "Dimensionality reduction of transition matrices". BFL = "difference from median". Additionally, the term "flow" could be misleading, as it implies a continuous and smooth transition. However, the method relies on discrete clusters of behaviors, and the transitions between these clusters might not always be smooth or continuous.

To summarize the above points, my main concern is whether the proposed framework adds enough benefit (on top of the need to run clustering approaches) to justify the increased complexity.

Reviewer #3 (Remarks on code availability):

I took a quick look at the code & README, but I do not have an environment setup where I could run it at the moment.

Version 3:

Decision Letter:

8th Oct 2024

Dear Johannes,

I am pleased to inform you that your Article, "Analysis of behavioral flow resolves latent phenotypes", has now been accepted for publication in Nature Methods. The received and accepted dates will be August 16th, 2023 and October 8th, 2024. This note is intended to let you know what to expect from us over the next month or so, and to let you know where to address any further questions.

Over the next few weeks, your paper will be copyedited to ensure that it conforms to Nature Methods style. Once your paper is typeset, you will receive an email with a link to choose the appropriate publishing options for your paper and our Author Services team will be in touch regarding any additional information that may be required. It is extremely important that you let us know now whether you will be difficult to contact over the next month. If this is the case, we ask that you send us the contact information (email, phone and fax) of someone who will be able to check the proofs and deal with any last-minute problems.

Please note that *Nature Methods* is a Transformative Journal (TJ). Authors may publish their research with us through the traditional subscription access route or make their paper immediately open access through payment of an article-processing charge (APC). Authors will not be required to make a final decision about access to their article until it has been accepted. [Find out more about Transformative Journals](https://www.springernature.com/gp/open-research/transformative-journals)

If you are active on Twitter/X (and haven't already done so), please e-mail me your and your coauthors' handles so that we may tag you when the paper is published.

Best regards,
Nina

Nina Vogt, PhD
Senior Editor
Nature Methods

** Visit the Springer Nature Editorial and Publishing website at www.springernature.com/editorial-and-publishing-jobs for more information about our career opportunities. If you have any questions please click here.**

Open Access This Peer Review File is licensed under a Creative Commons Attribution 4.0 International License, which permits use, sharing, adaptation, distribution and reproduction in any medium or format, as long as you give appropriate credit to the original author(s) and the source, provide a link to the Creative Commons license, and indicate if changes were made. In cases where reviewers are anonymous, credit should be given to 'Anonymous Referee' and the source.

RESPONSE TO REVIEWER COMMENTS

Reviewer 1:

Remarks to the Author:

In the current manuscript the authors tackle an important problem in the field of behavioral biomedical research of rodents, that is the comparison of complex behaviors and rich behavioral profiles across experimental groups and experiments. They introduce two dimensionality reduction pipeline termed Behavioral Flow Analysis (BFA) and subsequent Behavioral Flow Fingerprinting (BFF) for the assessment of group effects (BFA) and individual variability (BFF) in behavioral data sets obtained in open field test settings. Applying these approaches to different experiments and data sets, they demonstrate that these dimensionality reduction approaches are capable of detecting group and individual differences with higher sensitivity compared to previous methods. I believe that this will be an important methodological addition for the behavioral neuroscience field. I have a number of comments and questions, that I would like the authors to address or clarify.

RE: We thank the reviewer for the positive assessment of our work.

Comment 1: A major drawback of the current approach is the fact that all temporal information of the behavioral dynamics of the animals is lost. Open Field test situations are short and treated as individual units. This will be hampering application of these analysis pipelines, as specifically the change of behavior over time is a crucial feature in many experimental settings and research questions. Can the authors still implement this and test if BFA and BFF are capable to detect behavioral changes over time?

RE: We agree with the reviewer that the temporal dynamics within a task are important and we have added additional analyses to highlight this point. We wish to emphasize, however, that important temporal information is preserved in our analysis pipeline. First, we do retain temporal information within features (used for clustering) by incorporating information of the 15 leading and tailing frames, resulting in about a 1-second time window. Second, the whole analysis pipeline including BFA and BFF is based on transitions between clusters (i.e. behavior flow), again retaining temporal dynamics. In response to the reviewer's comment, we now added a time bin analysis to our BehaviorFlow package, allowing the user to split clustering results into smaller time bins (length of a time bin can be defined) and analyze them further (for more details, see Methods, page 42). To demonstrate the benefits and power of the time bin analysis, we applied it to two new datasets, which contain longer behavioral recordings (25-30 minutes). The first experiment mice were injected either with yohimbine (3 mg/kg, n=10) or vehicle (n=9) immediately before being tested for 30 minutes on a marble burying test (new Figure 6A, see below). We split each of these recordings into 6-minute time bins and analyzed the clustering results across bins. The findings of this analysis reveal striking temporal dynamics and are reported in the updated

Results section (page 24) and Figure 6. For ease of reading, the relevant section and the cropped figure is added below:

After visual inspection, we detected a single cluster (cluster M.3, Suppl. Video 15) that contained most of the digging behavior during the marble burying test and was strongly reduced in yohimbine animals, in agreement with the total number of buried marbles after the test (Figure 6E,F). We then leveraged the ability of our analysis pipeline to resolve temporal dynamics, and investigated how digging behavior evolves over time. Resolving the digging cluster over 6-minute bins, we see a slow onset of the strong drug effect over time (Figure 6G), consistent with drug uptake dynamics. We then used the BFF method to perform a 2D-embedding of binned transition data and identified similar temporal dynamics based on the analysis of the complete behavioral flow (Figure 6H). Interestingly, this method was also able to resolve more subtle behavioral changes within the control animals, which evolve over the 30-minute trial and likely capture habituation to the test environment.

Figure 6. BFA and BFF are transferable to other setups. (A) Schematic showing experimental design for marble burying test (MBT) after yohimbine or vehicle injection. [...] (E) Examples of final frames showing marbles buried after yohimbine (upper images) vs. vehicle injection (lower images). (F) Marbles buried differ significantly ($t(16)=16.2$, $p<0.001$) after vehicle vs. yohimbine injection. (G) Time spent in cluster M.3 (digging cluster) evolves differently over time for mice after vehicle or yohimbine injection. Each bin represents 6 minutes. (H) BFF shows change in behavior profile over time for yohimbine and vehicle. [...] p-values and adj. p-values are denoted as: * <0.05 , ** <0.01 , *** <0.001 . Error bars denote \pm SEM.

In the second experiment, we analyzed the recordings of the fear conditioning box during the inescapable footshock (IFS) paradigm (new Suppl. Figure 7F, see below). While the IFS animals during the first 5 minutes of habituation (Bin 1) show a very similar behavioral phenotype as control animals, the subsequent behavioral change due to the onset of random footshocks is very clearly resolved by the BFF analysis. The Results section (page 25) as well as Suppl. Figure 7 were updated to contain these findings, the relevant parts are presented below:

We also investigated behavioral changes over 5-minute bins in the fear conditioning box using BFF (Suppl. Figure 7J). The 2D-embedding clearly highlights the very strong behavioral change induced in the IFS group after onset of the random shocks (Bins 2-5).

Supplementary Figure 7. [...] (F) Schematic showing experimental design for exposure to fear conditioning box. [...] (J) BFF applied to fear conditioning box data. Each bin represents 5 minutes. [...] p-values and adj. p-values are denoted as: * <0.05 , ** <0.01 , *** <0.001 . Error bars denote \pm SEM.

For both datasets, our analysis pipeline is capable of capturing changes in behavior over time. We thank the reviewer for the suggestion, because adding a time bin analysis clearly strengthens our pipeline.

Comment 2: A second limitation is in my view the limitation to individual animals in the sparsest of environments (empty open field), which drastically limits the behavioral repertoire of the animals. Implementing multi-animal testing, which will include the animals' social behavior, as well as testing in more complex environments, would significantly improve the usability of these analysis pipelines.

RE: We agree that the behavioral repertoire in the open field test is limited. To show that our pipeline can readily be applied to a wide range of tests, we added several experiments conducted in more complex environments to the updated manuscript: 1) The marble burying test, 2) the light-dark box test, and 3) the fear conditioning box (see Methods, pages 31-32). Due to the new testing setups, we trained a new pose estimation network using DeepLabCut and conducted a new clustering for each experiment (see Methods, page 37). The results for these three experiments are reported in the updated Results section (pages 24-25) as well as the updated Figure 6 and Suppl. Figure 7. For ease of reading, these changes are inserted below:

Different behavioral tests. While we were able to show robust application of clustering followed by BFA and BFF analysis to multiple experimental datasets sampled in a homogenous recording and test setup, it is unclear how well this method applies to different behavioral tests, setups and labs. To reduce the number of animals used, we thus applied our analysis pipeline to existing video recordings of three behavioral tests from our lab. Top-view video tracking for all three experiments was again performed using DeepLabCut. Because our cluster classifier was not designed for stabilizing clusters across different setups, video framerates and behaviors, we again performed k-means clustering for each experiment, followed by the same analysis pipeline for BFA and BFF as described above.

Marble burying test. Mice were either injected (*i.p.*) with yohimbine (3 mg/kg) or vehicle (Figure 6A), immediately before exposure to the marble burying test. The floor of a rectangular cage was covered with wood chip bedding with marbles placed on top, which elicits digging behaviors that are not observed in the open field test. We again used 25 clusters as higher cluster numbers did not increase power (Figure 6B). Several clusters differed significantly between yohimbine and vehicle injections (Figure 6C), an effect that was confirmed by BFA (Figure 6D). After visual inspection, we detected a single cluster (cluster M.3, Suppl. Video 15) that contained most of the digging behavior during the marble burying test and was strongly reduced in yohimbine animals, in agreement with the total number of buried marbles after the test (Figure 6E,F). We then leveraged the ability of our analysis pipeline to resolve temporal dynamics, and investigated how digging behavior evolves over time. Resolving the digging cluster over 6-minute bins, we see a slow onset of the strong drug effect over time (Figure 6G), consistent with drug uptake dynamics. We then used the BFF method to perform a 2D-embedding of binned transition data and identified similar temporal dynamics based on the analysis of the complete behavioral flow (Figure 6H). Interestingly, this method was also able to resolve more subtle behavioral changes within the control animals, which evolve over the 30-minute trial and likely capture habituation to the test environment.

Light-dark box test. Mice were tested in the light-dark box test two days after exposure to chronic restraint stress (Suppl. Figure 7A). Power analysis showed that 50 clusters yielded the highest power in detecting group differences (Suppl. Figure 7B). While no cluster difference survived multiple testing correction (Suppl. Figure 7C), BFA did reveal a clear group effect (Suppl. Figure 7D), and a 2D-embedding showed a subtle phenotype (Suppl. Figure 7E).

Fear conditioning box. We tracked the animals' behavior in the fear conditioning box during the 25 minute IFS shock session (Suppl. Figure 7F). Our analysis pipeline revealed a very strong behavioral change in IFS animals using power analysis, cluster usage and BFA (Suppl. Figure 7G,H,I). We also investigated behavioral changes over 5-minute bins in the fear conditioning box using BFF (Suppl. Figure 7J). The 2D-embedding clearly highlights the very strong behavioral change induced in the IFS group after onset of the random shocks (Bins 2-5).

Figure 6. BFA and BFF are transferable to other setups. (A) Schematic showing experimental design for marble burying test (MBT) after yohimbine or vehicle injection. (B) Power analysis comparing different numbers of *k*-means clusters with the number of "marbles buried" for MBT. (C) Cluster occurrences in MBT. (D) BFA reveals a treatment effect of yohimbine (percentile=99.9, $z=7.1$, $p<0.001$, $d=6.57$) in MBT. (E) Examples of final frames showing marbles buried after yohimbine (upper images) vs. vehicle injection (lower images). (F) Marbles buried differ significantly ($t(16)=16.2$, $p<0.001$) after vehicle vs. yohimbine injection. (G) Time spent in cluster M.3 (digging cluster) evolves differently over time for mice after vehicle or yohimbine injection. Each bin represents 6 minutes. (H) BFF shows change in behavior profile over time for yohimbine and vehicle. [...] *p*-values and adj. *p*-values are denoted as: * <0.05 , ** <0.01 , *** <0.001 . Error bars denote \pm SEM.

Supplementary Figure 7. (A) Schematic showing experimental design for the light-dark box (LDB) test after chronic restraint stress (CRS). (B) Power analysis comparing different numbers of *k*-means clusters with classical readouts ("transitions" and "time in light") for the LDB test. (C) Cluster occurrences in LDB after CRS. (D) BFA reveals a treatment effect of CRS (percentile=99.5, $z=3.06$, $p=0.001$, $d=0.91$) in LDB. (E) BFF applied to LDB data after CRS. (F) Schematic showing experimental design for exposure to fear conditioning box. (G) Power analysis comparing different numbers of *k*-means clusters for fear conditioning box. (H) Cluster occurrences in fear conditioning box. (I) BFA shows a treatment effect of the fear conditioning box (percentile=99.9, $z=10.6$, $p<0.001$, $d=5.44$). (J) BFF applied to fear conditioning box data. Each bin represents 5 minutes. [...] *p*-values and adj. *p*-values are denoted as: * <0.05 , ** <0.01 , *** <0.001 . Error bars denote \pm SEM.

These additions show that our analysis pipeline can be applied to many different testing setups and can reveal treatment effects and resolve even subtle behavioral phenotypes. These additions considerably increased the scope of our manuscript and we thank the reviewer for the comment.

Concerning the comment about social interactions, it would be possible to track multiple animals, cluster their (social) behavior and apply our analysis pipeline afterwards. However, the vast majority of behavior screening in basic research and at pharmaceutical companies is conducted in single animals. We wanted to expand "personality profiling" of single animals, and the addition of a social conspecific complicates this by introducing behavioral variability of a second animal, adding noise to the system. We agreed with the editor that including social interactions exceeds the current scope of the manuscript. However, studying social interactions is an important and vibrant research area, thus we have updated the Discussion section (page 30) to point the reader to two elegant recent approaches that provide tools to track and analyze social behavior (Nilsson et al, 2022, BioRxiv; Bordes et al, 2023, Nature Communications):

Our work focused on tool development for an in-depth behavioral analysis of single animals, however we did not measure social interactions in this set of experiments. Two unsupervised approaches have recently emerged that provide a versatile platform for analyzing social interactions (Nilsson et al. 2020; Bordes et al. 2023). The tracking data from these tools is compatible with our analyses, thus our pipeline adds to the rapidly expanding toolbox for nuanced behavioral analyses of single or multiple animals.

Comment 3: When comparing the effectiveness of BFA to other methods, the authors rely on the p-values (e.g., Fig 1J, K, L), which in my view is not entirely correct. While a p-value is informative whether or not an effect exists, it will not reveal the size of the effect. The authors should therefore rather use effect sizes for these comparisons.

RE: We appreciate this helpful suggestion and we fully agree with the reviewer. In response to this comment, we have re-visited and re-written the entire section about power calculations. While power calculations in behavioral neuroscience are typically performed using a single behavioral measure from a given test (e.g. "distance moved" in the open field test), we had to devise an approach that can perform an effect size estimation and a power calculation based on the entire behavioral flow profile as assessed by our BFA approach. To achieve this, we introduce a new behavioral flow likeness (BFL) score, that computes the similarity of each individual animal to the median behavioral profile of all the animals in each of the two groups, in high-dimensional space (see Methods, page 40-41). The resulting BFL-score allows us to compare groups and compute the effect size (Cohen's d) and perform a power analysis based on the full complexity of the entire behavioral repertoire observed in a given test. The BFL method also proves useful for parameter optimization of the clustering approach, thus it is introduced in the updated Results section (page 6) as well as Figure 1 and Suppl. Figure 1. The relevant sections are presented below for ease of reading:

Behavioral Flow Likeness (BFL) for estimating effect size. *The ability of BFA to clearly resolve treatment effects prompted us to test whether analyzing the entire behavioral flow would increase statistical power. Power analyses are traditionally calculated based on a single behavioral measure, in the OFT typically "distance moved" or "time in center". For CSI, which is characterized by increased locomotion, "distance moved" yields more power than "time in center" (Figure 1K). To estimate the effect size between the high-dimensional*

behavioral flow profiles of two groups, we developed a behavioral flow likeness (BFL) score, which allows us to compare each animal's behavioral profile to the median of the two groups. Based on these BFL scores, we computed the effect size of the CSI treatment using Cohen's *D* (see Methods and Figure 1J). A subsequent power analysis revealed that the BFL-based approach yields more power than "time in center", but less than "distance moved" (Figure 1K). However, picking the single transition with the lowest adj. *p*-value (termed "best predictor for BFA") yielded a higher power compared to "distance moved" (Figure 1K). Thus, the BFL approach offers an unbiased estimation of effect sizes and enables power calculations based on the entire behavioral profile of an animal.

We next tested if the performance of our analysis approach in detecting group differences could be improved by systematically changing the number of clusters or the temporal integration period. Different integration periods (from +/- 5 frames to +/- 30 frames) showed only minor differences in power, but our initial choice of +/- 15 frames performed best (Suppl. Figure 1G). In contrast, a systematic comparison of cluster numbers (from 10 to 100) revealed that BFL based on 25 clusters yields the highest power (Figure 1L). Similarly, 25 clusters yielded by far the largest statistical power when using sensitivity assays based on *p*-values, where group sizes were successively reduced *in silico* (Suppl. Figure 1H).

Figure 1. Behavioral flow analysis increases power to detect phenotypes. [...] (J) Schematic of computing the behavioral flow likeness (BFL) score to estimate effect sizes. (K) Power analysis comparing classical readouts ("distance moved" and "time in center") to analysis of cluster transitions. (L) The number of clusters influences power to detect treatment effects. (M) Power analysis comparing three different clustering algorithms. [...]

Supplementary Figure 1. [...] (G) Power analysis comparing different integration periods. (H) BFA enhances sensitivity using various numbers of *k*-means clusters. [...]

For the remainder of the datasets, we also replaced all the "sensitivity assays" with power analysis curves (see updated Figure 2H,L,P and Figure 4H,L), and updated the Results sections accordingly. In addition, we used the power analysis curves to choose the number of clusters for the newly added datasets (see previous Comment 2).

When comparing the statistical power to detect group differences using the BFL method to single classical readouts, we do see a reduction for some of the datasets (see for instance Figure 1K above). However, we do believe that using a composite measure to estimate effect size is preferable as it does not require any a priori assumptions about which behavioral readout could be most informative. We address this point in the updated Discussion section (page 28):

Based on the behavioral flow likeness (BFL) score, we provide a tool to estimate effect size, enabling power analyses on the entire behavior profile. While traditional power analyses are based on a single measure, the choice of this measure can be arbitrary and requires detailed prior knowledge about the expected outcome. Although we advocate for using the entire behavioral flow profile for an unbiased power calculation, it is also possible to pick the transition that best predicts group differences, which will maximize statistical power (see Figure 1K).

We believe that the new approach to perform an estimation of effect sizes and power calculation - which is now integrated into our analysis pipeline - will aid in experimental planning (calculating required group sizes) and can lower the number of animals used in behavioral testing. This is in line with 3R principles (reduce, refine, replace) and can have a measurable impact on animal welfare.

Comment 4: In the BFF analyses (e.g., Fig. 3 G, H, I) there is still a large overlap between the groups and the directionality of change is not unidirectional. In 3H, for example, there seem to be subgroups of AS (45 min) mice that cluster together and are quite distinct. Do these subgroups of animals also show distinct behavioral profiles, potentially indicating different coping strategies in response to the stress exposure?

RE: Following the suggestion of the reviewer on the acute swim (AS) experiment, we stratified these mice into responders and non-responders using our updated analysis pipeline, including the previously introduced behavioral flow likeness (BFL) score (Reviewer 1, Comment 3). This was influenced also by a very helpful comment of Reviewer 2, who pointed out that statistical analysis should not be based on UMAP embeddings (Reviewer 2, Comment 5). Thus, we replaced the old method using UMAP embeddings with the new BFL-score, which can be interpreted as a continuous "treatment-responsivity measure". In the case of acute swim stress, animals showing a BFL score similar to the control group can be classified as non-responders, while the other mice can be classified as responders to the stressor (see new Figure 5G).

Figure 5. BFF captures individual variability and allows behavioral predictions. [...] (G) Schematic showing stratification of non-responding and responding groups based on behavioral flow likeness (BFL) score. [...]

Applying this new stratification method to the AS animals results in the following pronounced grouping of responding and non-responding animals:

BFF, applied to animals from AS dataset (controls: n=15, AS: n=15):

BFF showing individual behavioral differences over several different datasets:

Although we do not use 2D embeddings anymore to stratify animals into different subgroups, the BFF does reveal that our new approach seems to capture relevant behavior, which discerns different phenotypes as a response to a mutual stressor. Comparing responders and non-responders using BFA reveals a significant difference between their behavior flows (percentile=99.8, $z=4.14$, $p<0.001$, $d=3.63$). Specific clusters that significantly (adj. $p<0.05$) differ between the two groups are cluster 1 and 11 (supported rearing, see Suppl. Table 1) as well as cluster 7 (turning away from wall after a supported rear, see Suppl. Table 1).

BFA and significantly different cluster occurrences:

Animals that show a pronounced response to the acute stressor therefore do show a reduced number of supported rears, indicating a reduction in exploratory behavior.

Although these are intriguing results showing the possibilities of our updated analysis pipeline, we did not include them in the updated manuscript as this would have meant substantial changes to the structure and flow of the manuscript, because we turn to group stratification only once we introduce the inescapable footshock stressor (IFS) in Figure 5. We think that the IFS dataset is much better suited for showcasing the power of the BFL approach for group stratification, as it provides the possibility to study the predictive power of our stratification (higher group numbers and an experimental design dedicated to testing behavior pr.

Comment 5: When comparing flow diagrams (e.g. Figure 4D) the authors claim that these appear highly similar. Can they provide a quantification of this similarity? Making sense of these flow diagrams by visual inspection is currently difficult, due to the multiple transitions between clusters. It might be helpful to implement a threshold and omit cluster transitions that occur with a low likelihood.

RE: These are two helpful comments. To make the qualitative comparison easier for the reader, we replaced the old flow diagrams with network graphs (omitting transitions that occur only rarely, see new Figure 4D).

Figure 4. Clustering is transferable to new datasets with the same experimental setup. [...] (D) Comparison of average behavioral flow in control animals reveals a similar pattern between original clustering (CSI, AS and yohimbine) and transferred clustering (CRS and DREADD). Only transitions with an average appearance > 5 are shown. [...]

To quantitatively demonstrate the similarity of these behavioral flow profiles, we ran a BFA comparing the controls from the datasets used in the original clustering and the transferred clustering. BFA showed no significant differences (percentile=88.71, $z=1.19$, $p=0.117$, see plot below), indicating that the behavioral flow of these two groups is very similar.

Comment 6: When testing how well the analysis pipeline can be transferred to new datasets, the authors pick two clusters (1 and 11), which seem to describe similar behaviors. However, it is in my view not sufficient and convincing to only pick 2 out of 25 clusters. For the authors to conclude reproducible clustering transfer for completely new datasets, this needs to be shown and quantified for all 25 clusters.

RE: The reproducible cluster transfer is already addressed in part by our response to the previous comment (Reviewer 1, Comment 5). Further, we achieve a ~0.9 F-Score across all clusters when we train a cluster classifier and use appropriate cross-validation to estimate the accuracy of the classifier (Figure 2D). This cross-validation numerically assesses how often the transferred clustering works correctly. Finally, we show experimentally that three similar manipulations of the noradrenergic system (acute stress, DREADDs and yohimbine) lead to highly similar transition profiles across all transitions (in a 2D-embedding, Figure 4O). In response to the reviewer's comment, also we added a table describing the observed behavior in each cluster in datasets used for the original clustering and the transferred clustering (see new Suppl. Table 1). To generate this table, one behavioral expert was blinded and scored all videos representing the individual clusters (from both original and transferred clustering). These qualitative descriptions are very similar, further strengthening the conclusion that the behaviors represented in the original and the transferred clustering are very similar.

Supplementary Table 1. Manual inspection of clusters pre- and post-transfer

Cluster #	Original Clustering	Transfer Clustering
1	Supported rearing	Supported rearing
2	Movement along the wall	Movement along the wall
3	Center exploration	Center exploration
4	Locomotion in periphery	Locomotion in periphery
5	Clockwise turn	Clockwise turn
6	Exploration/sniffing of the wall	Wall approach and sniffing/exploration

7	Turning away from wall	Turn/move away from wall
8	Slow walk along the wall	Movement along the wall
9	Grooming and unsupported rearing	Grooming and some unsupported rearing
10	Locomotion	Movement in periphery
11	Supported rearing	Supported rearing
12	Slow rearing (supported and unsupported)	Unclear, in corners (very few examples)
13	Movement to/investigation of the corners	Sniffing/exploration of the wall
14	Transition from being stationary to locomotion	Movement towards the center, movement onset
15	Stationary exploration	Unclear (very few examples)
16	Counter clockwise turn	Counter clockwise turn
17	Slow and partial counter clockwise turn in corner	Corner exploration and some supported rearing
18	Unclear, slow exploration (sniffing, walking) some slow rearing	Unclear, sniffing and some slow rearing
19	Fast locomotion along the walls	Locomotion along the wall
20	Small stationary movements (including left/right sniffing)	Stationary with sniffing
21	Approaching the wall	Locomotion from center to periphery
22	Unclear	Unclear locomotion in periphery
23	Unsupported rearing and sniffing (low rearing)	Sniffing + unsupported rearing (few examples)
24	Sniffing, grooming with counter clockwise turn	Slow counter clockwise turn
25	Stop and go, S curve walking	Stop and go, S curve walking (left-right alterations)

Reviewer 2:

Remarks to the Author:

"Analysis of behavioral flow resolves latent phenotypes" by von Ziegler et al. presents a method of identifying behavioral clusters and analysing their transitions. The core idea of the paper is that by comparing the distance between a vector of transition probabilities rather than the probabilities themselves individually, the power to detect group differences in behavior is increased.

The authors make several claims about the advantages of their method, but these are not well supported. The broader claims about how this method fits with previous behavior analysis methods is also not sufficiently justified and in some places contradictory.

RE: We appreciate the critical feedback and hope to have addressed the reviewer's concerns in our answers below.

Main claims

1) Multiple testing in high-dimensional behavior analysis reduces power.

Comment 1: The authors show that this is true on their data using their representation of behavior and a 'classic' representation of open field data. This supports the step of their analysis where the combine transition probabilities into a single distance measure. However, success at this step doesn't support their representation itself. For example, you could apply this step to any behavior representation that includes transitions between behaviors (which is most representations of behavior). It should therefore be made clearer that the potential advantages of this step do not necessarily depend on the rest of the analysis and could equally be applied to, say, MoSeq.

This is really the central claim of the paper (the other two claims are not justified). However, to demonstrate its usefulness, it would be best to compare it to other ways of comparing transition matrices such as divergence statistics.

RE: The reviewer is correct that our analysis pipeline could equally be applied to other approaches, in fact it is specifically designed to work with input from various behavior segmentation approaches. To demonstrate this, we explicitly applied our behavioral flow analysis (BFA) to the output generated from two leading methods, VAME and B-SOiD (Suppl. Figures 2-3), and showed that these approaches - together with k-means - all produce very similar results. Moreover, addressing additional comments as well (Reviewer 2, Comment 2 and Reviewer 3, Comment 3), we added a series of new analyses where we applied VAME to an entire series of experiments, again showing that we can reproduce findings from k-means (Results section, page 18 and new Suppl. Figure 5):

Using the analysis pipeline with a different clustering approach. Thus far, we used k-means to cluster behavioral recordings into behavior motifs. To test if our analysis pipeline (Figure 2C) produces comparable results using a different clustering approach, we re-analyzed all five datasets using VAME for generating 25 clusters. Although the number of

significant clusters and transitions differed slightly from the *k*-means clustering, BFA reproduced significant group differences for all experiments (Suppl. Figure 5A-E), and the approach tended to increase power for detecting group differences compared to the standard OFT readouts. The 2D embedding again showed that all acute stressors (acute swim, yohimbine and DREADD) induced a similar phenotype, while the chronic stressors (CSI and CRS) shifted away from the control groups in the opposite direction (Suppl. Figure 5F). This shows that - even though the VAME clustering did resolve different behavioral motifs compared to *k*-means clustering (Suppl. Figure 5G) - our analysis pipeline is robust towards the choice of clustering approach and to the represented behavioral motifs.

Supplementary Figure 5. Results of our analysis pipeline applied to 25 VAME clusters (V.0-V.24) for (A) CSI (BFA: percentile=99.9, $z=6.99$, $p<0.001$, $d=1.26$), (B) Acute swim (BFA 45 min: percentile=99.9, $z=4.77$, $p<0.001$, $d=1.78$; 24 h: percentile=81.42, $z=0.83$, $p=0.203$), (C) yohimbine (BFA: percentile=99.9, $z=5.55$, $p<0.001$, $d=3.07$), (D) CRS (BFA: percentile=99.8, $z=3.87$, $p<0.001$,

*d=1.34) and (E) DREADD (BFA: percentile=99.8, z=4.65, p<0.001, d=1.66). (F) Using the 25 VAME clusters, BFF embeddings across all five experiments (CSI, acute swim, yohimbine, CRS and DREADD) reveal a separation of different behavioral phenotypes. The crossbars represent the average UMAP1 and UMAP2 values with SEM for each group. (G) Mapping 25 k-means clusters to 25 VAME clusters. p-values and adj. p-values are denoted as *<0.05, **<0.01, ***<0.001. Error bars denote \pm SEM.*

We chose k-means for the main part of the manuscript as it offers a simple and computationally efficient clustering algorithm, and it provides sufficiently detailed input for our downstream analysis pipeline. However, to emphasize to the reader that our analysis pipeline works with any clustering algorithm, we added the following text to the Results section (page 7):

Overall, we demonstrate that our analyses based on behavioral flow provide unbiased end-to-end methods for detecting general group differences (BFA) and estimating effect sizes (BFL). These methods demonstrably work with outputs from different established clustering algorithms. BFA and BFL provide a powerful basis for benchmarking available clustering algorithms and selecting optimal hyperparameters.

2) Transferring clusters between experiments can be a challenge that is solved using a 'classifier-in-the-middle' approach.

Comment 2: As I understand it, the authors train a neural network classifier to predict cluster membership using k-means clusters as the ground truth. The output from k-means is a point in space that defines a cluster. Points are assigned to a cluster based on which cluster 'center' they are closest to. The ground-truth best cluster to assign a point to is therefore the closest one. Because the closest cluster center to a new point can just be calculated directly, this direct approach will always be more accurate (it is already perfect according the setup of the problem), I don't see what the neural network adds here. In places where the neural network disagrees with the simple procedure, it is basically by definition because the neural network has made a mistake.

Furthermore, this also shows that the premise of claim one is not really correct. For a given set of clusters, a new point from a different experiment can be assigned quickly and easily. There is an issue with clusters changing fundamentally from one experiment to the next, but the 'classifier-in-the-middle' doesn't change that. As the authors do themselves in the last section, one way to address this problem is just to redo the clustering on the new data. Again, the classifier is unnecessary.

In case I've completely missed the point, the authors should at least consider making the explanation of the training and ground truth clearer.

RE: The reviewer correctly points out that if k-means clustering is used, the classifier-in-the-middle approach is not necessary. We appreciate the feedback, as we should have made our reasoning for this clearer: There are two main reasons why we opted for a classifier-in-the-middle

approach. First, there are two major normalization steps we perform on the feature data. For each animal, we normalize data so that differences between animals (e.g. larger vs. smaller animal) are corrected for. Then we normalize across all samples in each experiment so that population boundaries and features are properly represented, and all the features are equally weighted for k-means clustering. The clustering will split this normalized data on its k-means logic, however we then train the classifier on the ground-truth with non-normalized data (see also Methods, page 38). While we could circumvent this by saving the exact metrics for the normalization scheme, there is a second, more important reason for the classifier-in-the-middle: As we show in Suppl. Figures 2, 3 and 5, and as this reviewer pointed out in Comment 1, the clustering/segmentation data could come from any source, e.g. from a non-linear embedding such as HDBscan where embedding of new data needs access to the old data. Our classifier provides an efficient solution for this, adding great versatility to our pipeline. We extended the Results section (pages 11-12) to explain this reasoning:

To obtain comparable clustering results across these experiments, all the data would need to be used in one big clustering experiment. However, depending on the size of experiments, this can become computationally very expensive. Further, although k-means allows calculating which cluster center is closest to a new data point, other clustering algorithms (for instance density-based clustering as applied by B-SOiD) would need access to the original clustering data to embed new behavioral recordings. To offer a computationally efficient solution for every choice of clustering algorithm, we stabilized clusters using a classifier-in-the-middle approach.

We also incorporated a whole new paragraph into the Results section, showing that our whole analysis pipeline - including the classifier-in-the-middle approach - can be applied to clustering results other than k-means. These changes can be found in the reply to the previous comment (Reviewer 2, Comment 1).

3) The problem of individual differences

Comment 3: Individual differences are indeed prevalent in behavior data. However, the proposed solution of considering individual behavioral fingerprints in a high-dimensional space is not unique to the current method. Placing individuals in a high-dimensional space for comparison can be applied to any behavioral fingerprinting approach (quantitative fingerprinting from tracking data dates back at least to 2002 <https://pubmed.ncbi.nlm.nih.gov/12191753/>).

RE: We do not claim that the idea to represent behavior in high-dimensional space (behavioral flow fingerprinting, BFF) is per-se novel, but that we provide a clear framework for how a series of approaches can be implemented. The addition of our "classifier-in-the-middle" allows BFF to compare high-dimensional data across experiments, thus the demonstration that various pharmacological, environmental and circuit neuroscience manipulations can be compared directly in one major 2D-embedding (spanning years of work from our laboratory) (Figure 5F) is a unique asset of our approach. The new addition of the "behavioral flow likeness" (BFL) metric provides

an additional, novel way to interpret individual differences, by assessing how similar a given animal is to either of the treatment groups (see also response to Reviewer 2, Comment 5). This approach enables predictions about individual behavior in future tests, as demonstrated in the newly analyzed Figures 5G-L (where fear extinction after footshocks is predicted by behavior in the open field test), and as described for acute swim stress in the response to Reviewer 1, Comment 4.

Comment 4: Furthermore, the basic claim that their method improves statistical power relies on ignoring these details (at the expense of interpretability). It is a bit of a contradiction to claim the same method improves power by combining transitions into a single measure and also improves interpretability by not combining transitions into a single measure.

RE: We understand the reviewer's point, but we do not see a contradiction here, because our pipeline is composed of a series of sequential analyses. A central idea behind our approach is that BFA is a first-pass analysis (like in an ANOVA) to determine whether there is a difference worth exploring in the first place. This offers a solution to increase power in big-data behavior screens, which otherwise suffer from dramatically reduced power if multiple testing corrections are appropriately applied (a fact that has rarely been admitted in the literature thus far). We go to great lengths to show that this BFA approach is sensitive and can detect subtle group differences, and we have extended our pipeline to also report effect sizes and power estimates (see Reviewer 1, Comment 3). If the first level of analysis (BFA) gives permission to go forward, "behavioral fingerprinting" becomes permissible. Thus, there is no contradiction between a first-pass analysis to test overall significance, followed by an in-depth analysis (of individual behavioral clusters or transitions) only if a significant group effect is observed.

Comment 5: In terms of how the fingerprinting approach is implemented here, it is not recommended to do statistics on UMAP dimensions. Reducing the dimensionality down to 2 and then quantifying the mean and standard error can lead to difficulties in interpretation because different projections can give very different results. Repeating the UMAP with different seeds is insufficient to address this issue. See <https://journals.plos.org/ploscompbiol/article?id=10.1371/journal.pcbi.1011288> for some more extreme examples.

RE: We thank the reviewer for highlighting this weakness, we agree that UMAP dimensions are not suitable for stratifying animals into different groups. We have now replaced this approach with a solution that does not rely on UMAP dimensions and is completely linear. This "behavioral flow likeness" (BFL) score computes the similarity of each individual animal to the median behavioral profile of all the animals in each of the two groups, in high-dimensional space (see Methods, page 40-41, and new Figure 1J).

Figure 1. Behavioral flow analysis increases power to detect phenotypes. [...] (J) Schematic of computing the behavioral flow likeness (BFL) score to estimate effect sizes. [...]

The BFL score can then be used to estimate effect size using Cohen's D (see also Reviewer 1, Comment 3), but also as a treatment-responsivity measure. In the revised manuscript, we applied the BFL method to stratify animals exposed to inescapable footshocks (IFS) into responders and non-responders (new Figures 5G,H). The updated Results section (page 22) and Figure 5 incorporate all these new findings using our BFL approach:

Capturing stress-responsiveness in high-dimensional space. Similar to stress resilience in humans, there is great variability in stress responsiveness even within genetically identical mice³². Distinguishing animals whose behavior changes dramatically (responders) from those who are less affected by stress (non-responders), is often done using a single behavioral measure³³, which raises questions about bias, reliability and validity³⁴. To address this, we used the BFL score that, beyond its usefulness for computing effect size, can be viewed as a continuous "treatment-responsivity measure", which contains information about the behavioral change for each individual mouse. Thus, IFS-exposed mice with a BFL score similar to the control group were classified as non-responders, while animals outside the range of BFL scores of the control group were classified as responders (schematic shown in Figure 5G, grouping in Figure 5H). Using this new group assignment, we compared OFT2 performance and found - as expected - different cluster representation and behavioral flow (Figure 5I,J) and a very strong group difference using BFA (Figure 5K). More interestingly, this group assignment (based on OFT2 performance) allowed a forward prediction, revealing that responders showed a stronger and more protracted freezing response during 6 consecutive days of extinction training (Figure 5L).

Figure 5. BFF captures individual variability and allows behavioral predictions. [...] (G) Schematic showing stratification of non-responding and responding groups based on behavioral flow likeness (BFL) score. (H) $\log(\text{BFL})$ scores for control, non-responder and responder animals. (I) Cluster occurrences in non-responding vs. responding mice. (J) Absolute differences in behavioral flow in responding vs. non-responding mice. (K) BFA reveals a group effect between responding and non-responding mice (percentile=99.9, $z=6.01$, $p<0.001$, $d=3.34$). (L) Freezing response during extinction sessions for control, non-responder and responder mice. p -values and adj. p -values are denoted as: * <0.05 , ** <0.01 , *** <0.001 . Error bars denote \pm SEM.

Furthermore, we applied the same pipeline to stratify acutely stressed animals into responders and non-responders to answer the comment raised by another reviewer (Reviewer 1, Comment 4). We found a change in the frequency of supported rears (represented by a sequence of clusters 1, 7 and 11) between responders and non-responders, showing again that the BFL method is suitable to distinguish subtle differences in the behavior profile of individual animals. We thank the reviewer for this comment as we believe the addition of the BFL strengthens the revised manuscript.

Specific points:

Comment 6: "The majority of preclinical biomedical research is conducted in mice". Probably more accurate to say that the majority of *in vivo* preclinical biomedical research is done in mice.

RE: Thank you, we have changed the first sentence in the Introduction (page 3) accordingly.

Comment 7: intro, first paragraph: "the advent of pose estimation" isn't necessarily the right transition to emphasise. For example, even with centroid tracking, high-dimensional fingerprinting of mouse behavior can be done. Kafkafi et al. have even shown that it can sensitively phenotype drug treated animals in open field assays. See <https://www.ncbi.nlm.nih.gov/pmc/articles/PMC8056474/> as well as their two earlier papers introducing the method in 2008 and 2009. These papers should be cited and discussed in the context of the new method.

RE: We thank the reviewer for pointing us to this literature. As suggested, we have included these references in the Introduction (page 3) to highlight the fact that open field behavior contains rich information about the phenotype and genotype of mice. We also point out that this algorithm was not widely adapted in the field, presumably due to the complexity of the manually defined and curated profiles of behavioral patterns.

Comment 8: "However, this would entail re-running the unsupervised clustering every time new data is added and soon become computationally challenging as previously observed". There are approaches to k-means that scale better with large datasets. In any case, as discussed

above, the existing clusters can be kept and new data assigned to the right cluster with little computation even for large datasets.

RE: This sentence was removed during revisions. As already addressed in a previous comment (Reviewer 2, Comment 2), we changed the corresponding Results section (pages 11-12) to address this issue.

Comment 9: "As a result, large group numbers are required, which runs counter to the goal that these approaches set out to achieve in the first place." Some methods of detailed phenotyping may demonstrate increased statistical power (and therefore justify a reduction in animals), but I wouldn't say reducing animal numbers is 'the', or even the main, goal of behavioral fingerprinting.

RE: We agree with the reviewer that there are many reasons why a detailed analysis of animal behavior is valuable and important. However, if advanced methods of analysis enable us to reduce the number of animals for basic research, it becomes an ethical responsibility for researchers to implement such tools. Following the 3R principles (replace, reduce and refine), researchers working with animals have to reduce animal numbers in their experiments while also reducing the amount of strain imposed. Our revised analysis pipeline offers a power analysis based on the whole behavior profile (see Reviewer 1, Comment 3 for more details), providing a tool for others to make informed decisions about future experiments following the 3R principles. We have re-written the sentence to incorporate the reviewer's feedback:

As a result, large group numbers are required, which runs counter to animal welfare regulations that call for a more refined analysis of rodent behavior that reduces the number of animals needed for testing.

Reviewer 3:

Remarks to the Author:

The paper aims to quantify behavioral differences between groups of mice under different conditions. The authors propose a pipeline that takes in behavior clusters (from existing approaches such as VAME, BSOID, etc.), then computes Manhattan distance between cluster transitions of the group means, and perform bootstrapping to estimate null distribution of group distances. The authors then use the right-tailed z-test to compare the actual distance to the null distribution to detect group differences. The authors name this pipeline behavioral flow analysis (BFA). Experiments are performed on a set of mouse datasets (CSI, acute swim, yohimbine) + one more from a different lab, to show that BFA is able to identify group differences across settings. The authors also propose to use this pipeline to measure individual animal differences as well (behavioral flow fingerprinting BFF). I think the paper tackles an important problem, but I have some questions about the approach and experiments:

RE: We thank the reviewer for this assessment and hope that we have addressed all concerns in our responses to this and the other reviewers.

Comment 1: The paper claims that the "classifier-in-the-middle" approach improves transferability in the introduction. However, the classifier still needs to be trained on clusters in each experimental setting in a supervised way. The authors note in the introduction that "it is difficult to transfer a model between different testing conditions, datasets or laboratories" and claims that the supervised classifier addresses this; however, since the supervised model also needs to be trained in each setting (similar to the clustering approach), it is unclear to me if the claim is true.

RE: The reviewer raises an important point about two different levels of transferability, "within-setup" and "across-setup" transferability. We show that we can achieve successful transfer of our analyses within a given test setup/lab across a wide range of cohorts and thus compare phenotypes across experiments (see Figure 2 and 4). While this is an important feature, the reviewer correctly points out that for "across-setup" transferability, this approach requires re-clustering and re-training, albeit just once. We highlight this in the updated manuscript (Results section, pages 24-25) when we show that our pipeline works on data gathered in different setups for different behavioral tests (updated Figure 6 and Supplementary Figure 7). We refer the reviewer to our response to Reviewer 1, Comment 2.

We also emphasize the implication that clustering results are not directly comparable across different laboratories, but require a new clustering. That said, the pipeline can be directly applied to data from a different laboratory (see updated Figure 6I-O and Suppl. Figure 7K-O, pasted below for ease of reading). We believe that transferring clusters directly between setups/labs is a remaining challenge that is unlikely to be resolved with any of the most advanced clustering approaches and is beyond the scope of our manuscript. One possible solution would be an open-source testing setup (hardware), so that the exact same setup can be used in different labs.

Figure 6. BFA and BFF are transferable to other setups. [...] (I) Schematic showing experimental design for OFT after yohimbine or vehicle injection in another laboratory using a different OFT setup. (J) Schematic showing experimental design for OFT after diazepam or vehicle injection. (K) Cluster occurrence in yohimbine, or (L) in diazepam. (M) BFA shows treatment effects after higher doses of diazepam (1 mg/kg: percentile=68.93, $z=0.5$, $p=0.308$; 2 mg/kg: percentile=99.9, $z=4.14$, $p<0.001$, $d=2.72$; 3 mg/kg: percentile=99.9, $z=4.91$, $p<0.001$, $d=4.2$). (N) BFA reveals behavioral changes after different doses of yohimbine injections (1 mg/kg: percentile=99.9, $z=4.44$, $p<0.001$, $d=4.32$; 3 mg/kg: percentile=99.9, $z=6.1$, $p<0.001$, $d=6.12$; 6 mg/kg: percentile=99.9, $z=6.37$, $p<0.001$, $d=6.33$). (O) BFF separates the different doses of diazepam and yohimbine injections. p -values and adj. p -values are denoted as: * <0.05 , ** <0.01 , *** <0.001 . Error bars denote \pm SEM.

Supplementary Figure 7. [...] (K) Comparison of classical behavior readouts for different doses of diazepam or yohimbine (time in center for diazepam: $F(3,28)=0.78$, adj. $p=0.514$; for yohimbine: $F(3,28)=3.2$, adj. $p=0.038$; distance moved for diazepam: $F(3,28)=4.09$, adj. $p=0.026$; for yohimbine: $F(3,28)=46.27$, adj. $p<0.001$; supported rears for diazepam: $F(3,28)=4.68$, adj. $p=0.022$, for yohimbine: $F(3,28)=29.18$, adj. $p<0.001$; unsupported rears for diazepam: $F(3,28)=8.85$, adj.

*p=0.001; for yohimbine: $F(3,28)=4.02$, adj. $p=0.021$). (L) Power analysis comparing different numbers of k -means clusters with classical OFT readouts after treatment with diazepam or (M) yohimbine. (N) BFA reveals differences between higher doses of diazepam (2 or 3 mg/kg) compared to lower doses (1 mg/kg) (1 vs. 2 mg/kg: percentile=96.4, $z=2.04$, $p=0.021$, $d=0.19$; 1 vs. 3 mg/kg: percentile=99.7, $z=3.34$, $p<0.001$, $d=1.15$; 2 vs. 3 mg/kg: percentile=49.45, $z=-0.1$, $p=0.541$). (O) BFA shows treatment differences between different doses of yohimbine (1 vs. 3 mg/kg: percentile=99.7, $z=3.71$, $p<0.001$, $d=2.45$; 1 vs. 6 mg/kg: percentile=99.8, $z=5.68$, $p<0.001$, $d=5.36$; 3 vs. 6 mg/kg: percentile=99.2, $z=3.43$, $p<0.001$, $d=1.49$). p -values and adj. p -values are denoted as: * <0.05 , ** <0.01 , *** <0.001 . Error bars denote \pm SEM.*

Comment 2: Another question I have from the paper is the evaluation setup - it seems BFA almost always results in showing treatment effects (the only test with no effect is one experiment in Fig 5E). The number of negative tests is thus very small (let me know if I might have missed others in the paper). Have the authors done more checks on the method, for example, by taking two similar groups (both with/without treatment) to show that BFA also finds this to be the case?

RE: As the reviewer points out, we have shown “negative findings” in an experiment where an inescapable footshock stress (IFS) induces a very strong effect shortly after IFS, but we show that there was no group effect before stress exposure nor after 8 days of recovery from stress (Figure 5E). In addition, we have also shown that there is no treatment effect 24 hours after swim stress (Suppl. Figure 4), a finding that is consistent with our previous work, as animals quickly recover from this acute stressor. Further, the demonstration that BFA detects graded effect sizes that scale with drug dosage (yohimbine and newly added diazepam, Figure 6M,N and Suppl. Figure 7N,O) is strong evidence that BFA detects and adequately quantifies biologically meaningful changes.

However, the reviewer's comment prompted us to add an additional experiment to demonstrate already early in the manuscript that BFA does not "create" treatment effects when none are expected. Hence, we turned to the first experiment (chronic social instability, CSI), where we had a large sample size, and randomly divided the animals from the control group and the animals from the CSI animals into two subgroups ($n=14-15$ /group). We then compared these subgroups and found that - as expected - BFA does not detect differences between the two control groups, nor between the two CSI groups. This finding is now incorporated in the updated Results section (page 5, see new Suppl. Figure 1F):

To rule out that BFA arbitrarily generates effects, we randomly divided only control animals and only CSI animals into two subgroups ($n=14-15$ /group), and showed that BFA does not detect differences between the two control groups, nor between the two CSI groups (Suppl. Figure 1F).

Supplementary Figure 1. [...] (F) BFA shows no differences for only control (percentile=60.3, $z=0.2$, $p=0.419$) or only CSI animals (percentile=45.0, $z=-0.2$, $p=0.578$). [...]

Comment 3: Additionally, there lacks comparison of the proposed approach to baseline methods, so it is difficult to evaluate the significance of this pipeline. There is one comparison in Supplementary Figure 1D compared to using clustering or transition matrices alone; but the experiment is not comprehensive, since it is (1) only on one of the subsets of the data, (2) unclear if the clustering/transition matrices would perform better with different numbers of clusters or hyperparameters. Another factor that makes comparing the experiments difficult is that the authors have used different input clustering approaches in different data subsets as input to BFA - is there any comparison of which clustering approach should be used with BFA?

RE: We thank the reviewer for these constructive suggestions, which we have addressed directly:

(1) Throughout the paper we have several instances where we demonstrate that just looking at cluster numbers or transitions (with appropriate multiple-testing correction) does not reveal treatment effects, while our behavioral flow analysis (BFA) successfully does (e.g. for an acute swim stressor or for yohimbine treatment, see Figure 2 I,J and 2M,N).

(2) Addressing this comment as well as another comment (Reviewer 1, Comment 3), we incorporated power analyses throughout the revised manuscript comparing our approach to baseline methods such as single transitions (called “best predictor for BFA”) as well as classical behavioral readouts (for instance “distance moved” or “time in center” for open field tests). These power comparisons can additionally be used to adjust hyperparameters (e.g. number of clusters or integration period, see Figure 1L and Suppl. Figure 1G) and compare clustering algorithms (Figure 1M). All these comparisons can be found in the updated Results section (Figures 1, 2, 4, 6, and Suppl. Figures 1, 5 and 7). We also updated the Discussion section (page 28) to review our findings:

Based on the behavioral flow likeness (BFL) score, we provide a tool to estimate effect size, enabling power analyses on the entire behavioral profile. While traditional power analyses are based on a single measure, the choice of this measure can be arbitrary and requires detailed prior knowledge about the expected outcome. Although we advocate for using the entire behavioral flow profile for an unbiased power calculation, it is also possible to pick the transition that best predicts group differences, which will maximize statistical power.

(3) Regarding the difficulty of comparing experiments: we have addressed this point by applying the exact same processing pipeline to all experiments presented in the paper – which allows us to directly compare the results of all experiments in a single 2D embedding (Figure 5F). This is one of the core strengths of our approach, showing that experiments conducted in many different

cohorts can be compared and reveal meaningful patterns in the data that align with the underlying brain processes.

(4) The choice of input clustering approach is up to the user. We chose k-means because it's simple and efficient, but we show that the BFA pipeline works well with different algorithms (see Figure 1M and Supplementary Figures 2-3). In addition, addressing also a comment of another reviewer (Reviewer 2, Comment 1), we extended this comparison in the revised version and show that the alternative clustering algorithm VAME (used already in Suppl. Figure 2) produces very similar results to k-means when applied to the same sets of experiments. Our findings using VAME are incorporated in the updated Results section (page 18) and in the new Suppl. Figure 5. We refer the reviewer to our previous response (Reviewer 2, Comment 1).

Comment 4: While there are discussions on other works in clustering, other methods with similar goals/settings are not discussed (for example, contrastive analysis [A], representation learning ([B,C])). These methods also output representations that could be clustered. I don't think it's necessary to run experiments on everything as baselines, but more discussions of other works in these areas would be helpful to situate this method with respect to related works.

[A] Weinberger et al., Feature Selection in the Contrastive Analysis Setting

[B] Sun et al., Task Programming: Learning Data Efficient Behavior Representations

[C] Azabou et al., Learning Behavior Representations Through Multi-Timescale Bootstrapping

RE: We thank the reviewer for pointing us to this literature. As mentioned by the reviewer, the three manuscripts referenced above offer different attempts on constructing meaningful representations of input data. Although not shown specifically by [A], we do agree with the reviewer that all three methods could be applied to behavioral data for capturing relevant behavioral representations/motifs, which in turn could be used as input for various clustering algorithms. In fact, VAME (one algorithm we compare to k-means, see Suppl. Figure 2 and 5) does apply a similar technique before clustering: It uses a variational RNN autoencoder to embed pose-estimation data into a low-dimensional latent space before behavioral motifs are inferred using a Hidden-Markov-Model. We added the suggested references to the updated Discussion section (page 28):

We are in the midst of a revolution in behavioral neuroscience, where big data approaches allow us to describe and quantify rodent behavior in unprecedented detail^{4,17}. While some work focuses on identifying relevant representations and motifs from behavioral recordings (Weinberger et al. 2023; Sun et al. 2021; Azabou et al. 2022), others focus on identifying group differences by quantifying the occurrence of each cluster or the number of transitions between clusters^{12, 18,20,23,44,45}.

Nonetheless, we do want to point out that the goal of our work is not the inference of behavioral motifs nor how to best cluster them. We focus on providing a streamlined tool that makes the detection and post-hoc analysis of behavioral data more comprehensive and less biased towards single, manually-chosen behaviors.

Comment 5: Finally, this method relies on input clusters (which itself relies on estimated keypoints...). This is potentially a brittle pipeline, as it relies on many previous computation steps, some of which are hard to evaluate (for example, it is difficult to determine the "correct" number of clusters and clustering hyperparameters). Additional experiments, such as ablation studies on the cluster quality and cluster number / other hyperparameter variations, would help demonstrate the ability of this pipeline to be applied across different lab settings.

RE: We agree with the reviewer that changing hyperparameters or introducing noise to the system can influence the downstream analysis to various degrees. Concerning the first step of our workflow, pose-estimation does provide very robust and reliable body-point tracking, and has been widely adopted in the field. In regards to clustering hyperparameters like number of clusters or integration period, we do show in the revised manuscript that our pipeline produces robust results independent of the choice of these hyperparameters. When comparing different clustering algorithms (k-means vs. VAME vs. B-SOiD), we do see that we can reproduce group effects regardless of the chosen algorithm. For more details on these results, we do refer the reviewer to our previous response (Reviewer 3, Comment 3). Taken together, we explore several degrees of freedom in our updated manuscript where we show the robustness of our approach.

RESPONSE TO REVIEWER COMMENTS

Reviewer 1:

The authors thoughtfully addressed all of my previous comments and incorporated significant improvements in the revised version of the manuscript. With these changes and additions the manuscript has in my view significantly improved and I can now recommend publication. I believe that the presented analysis pipeline will be of high interest in the behavioral neuroscience community.

RE: We thank the reviewer for the positive assessment of our revised manuscript.

Reviewer 2:

The authors have addressed my specific concerns. I'm not aware of any remaining technical issue in the paper.

RE: We are happy that our revisions addressed all the previous concerns of the reviewer.

I realise the paper represents a lot of work and it seems like a good analysis pipeline, but it exists in the context of many other good analysis pipelines for behavior data. I don't see the kind of methodological advance I would expect for a paper in Nature Methods.

Specifically:

1) Clustering to identify states in behavioral data is now common and the authors use several published methods.

RE: We agree with the reviewer that clustering of behavioral data is implemented by more and more labs throughout the behavioral neuroscience field. As a result, post-clustering pipelines are increasingly needed. Our manuscript offers such an analysis pipeline which can be employed regardless of the testing and recording setup or clustering algorithm used (see also Comment 3 below).

2) The idea of comparing transition matrices is also not new even if the specific bootstrapping approach is. For example, this article uses transition matrices to compare behaviors

<https://doi.org/10.3758/BF03192788> and cites a tool written to compare behavioral transition matrices (https://brill.com/view/journals/beh/125/3-4/article-p157_1.xml) from the 90s.

RE: As mentioned by the reviewer, Hemerik et al. (*Behavior Research Methods*, 2006) use behavior transitions between 5 manually recorded behaviors (i.e. walking, grooming, standing still, flying, ovipositioning) to investigate differences in host-searching behavior of a parasitoid insect due to changes in plant species composition. To include this early work on behavior transitions, we added its reference to the updated Discussion section (page 12).

The software tool Matman 1.1 (from the company "Noldus Information Technology") used in the study is a commercial tool which implements different ethological analyses of matrices. More specifically, the study uses the Mantel's Z test (or just Mantel test) to assess similarities between two behavior transition matrices. Normally, this test is performed on two different distance matrices, for instance between genetic and geographic distances. Applied to our transition matrices (which are no distance matrices per se), this test will almost always show a strong correlation between two groups, therefore it is not suitable to detect treatment effects. However, it could be applied to compare two different clustering algorithms to assess their similarity. Although commercial tools do offer several advantages (e.g. client support, maintenance, user-friendly interfaces), we do think that our open source analysis pipeline provides a transparent, customizable tool that also gives scientists with less monetary resources a possibility to analyze their data. Furthermore, the rapid rate of progress in the field of behavior tracking and downstream analysis has recently outpaced commercial solutions (see e.g. Sturman et al, 2020: <https://pubmed.ncbi.nlm.nih.gov/32711402/>).

3) I understand why the authors use the classifier, but it still seems unnecessary, or at least a solution to a problem that could be solved another way more simply. As the authors say in their response, they could save the normalization metrics and share those to enable new data to be embedded. They then say that some clustering methods would require the full dataset to re-embed new data, but if comparing across experiments is a priority, the authors could solve this by sticking with k-means.

RE: It is true that we could remove the cluster classifier when using k-means by saving the normalization metrics and centroids of the different clusters. However, we do want to offer an analysis pipeline which works regardless of the clustering algorithm. This makes the implementation of our method much easier for others, as they do not have to swap their whole clustering pipeline to k-means. Instead, they can just take their clustered data and run our analysis pipeline on top. We consider this a key advantage of our pipeline and show its transferability to other clustering algorithms in Supplementary Figure 2, 3 and 4.

Reviewer 3:

I'd like to thank the authors for the detailed feedback to reviewers and updated manuscript with additional experiments. The paper is definitely stronger compared to the initial submission. The modifications include the behavioral flow likeness score, new discussions (ex: different clustering approaches), and additional experiments suggested by reviewers (ex: showing whether the method detects differences when there's none). I'm focusing my review on the new additions as well as comments/responses from other reviewers:

RE: We thank the reviewer for reading through our feedback and the revised manuscript, and for the positive assessment that all these revisions have strengthened our manuscript.

- Classifier-in-the-middle: I appreciate the clarifications from the authors on the classifier-in-the-middle approach, and I see now it is a supervised classifier trained on the cluster data (potentially from many sources). Since the classifier is trained in a supervised way, this approach inherits the limitations of supervised learning (ex: limited ability to generalize to new data, over-confidence on out-of-distribution data). To address this, the authors could assess the classifier's performance on different train/test settings (e.g., using different experiments as train/test), a wider range of datasets, and/or better define limitations and what constitutes a "new setup" requiring classifier retraining.

RE: The reviewer is right that the classifier-in-the-middle is supervised and therefore inherits the limitations of supervised classifiers. Generalization to new data is limited in these classifiers, which we show in Figure 2D, where we assess the performance of the cluster classifier for each cluster using cross-validation. With an average F1-score (= harmonic mean of precision and recall) of about 0.92, most frames do get the correct cluster assigned, but there are some remaining mismatches. To assess the influence of the chosen training dataset on this performance, we decided to run the following analysis: From the 6 experiments with OFT recordings performed in our lab (CSI, AS, yohimbine, CRS, DREADD, IFS), we selected 10 files at random (6 x 10 files = 60 files). We then performed a k-means clustering with 25 clusters on this new subset of data. Finally, a cluster classifier was trained on the recordings and their assigned clusters of 5 experiments, while the remaining experiment served as a validation dataset. Each experiment was left out once, and the results of this analysis are shown below:

These results show that the choice of training dataset (out of the datasets presented in the manuscript) only has a minor influence on the performance of the cluster classifier on validation data and that it generalizes well to unseen data.

Whether it is necessary to perform a new clustering and therefore train a new cluster classifier depends on several considerations:

- 1) If changes occurred in the test setup (e.g. different maze/arena) and/or recording setup (e.g. different camera angle, different frame rate)
- 2) If changes occurred in the feature set (e.g. different tracking points or features computed based on these tracking points) or the clustering algorithm
- 3) If the emergence of a new behavior (not represented in the training data) is expected

We specifically call the reader's attention to these considerations in the new "Limitations and Outlook" section of the revised Discussion.

- I read the author response on the comment about pipeline brittleness and I have remaining concerns on this point. The pose estimator, clustering algorithm, and BFA (or variants) each add more hyperparameters and steps to the analysis. For instance, the performance of the pose estimator can be affected by factors such as occlusion, lighting conditions, and individual differences in animal appearance. The clustering algorithm's performance depends on the choice of hyperparameters and the quality of the input features (as well as choice of pre-processing steps). Additionally, the BFA method itself relies on the accuracy and stability of the clusters identified in the previous steps. While the authors demonstrate their pipeline in different rodent experiments & with a few clustering algorithms, it's important to acknowledge that the framework might not generalize well to situations where tracking is unreliable or when the behavioral repertoire differs significantly from the training data (ex: if classifier-in-the-middle is used). Finally, simply because a method is widely used, doesn't mean it is reliable (especially under all distributions of data/behaviors). To address these concerns, I suggest the authors provide a more thorough discussion of the potential limitations and failure modes of their pipeline.

RE: We do agree with the reviewer that the whole analysis pipeline (including animal tracking, clustering features computed based on this tracking and then analyzing these clusters) rests on many (hyper-)parameter choices. In response to the reviewer's comment, we decided to study the influence of unreliable tracking on our behavioral flow analysis (BFA). We tested how removing different tracking points influences the detection of a treatment effect in the chronic social instability (CSI) dataset. We only ran this analysis for a subset of points, as a complete "feature selection" analysis would go beyond the scope of our manuscript. After removing a single tracking point, a reduced set of features was computed. Depending on the missing tracking point, different sets of features were removed (see Supplementary Table 2 for list of features and tracking points used to compute them). We then ran these sets of features through a new k-means clustering with 25 clusters. The BFA results of these new clusterings are shown below:

Without bodycentre (-10 features):	Without bodycentre left (bcl; -8 features):	Without nose (-7 features):
 $p=3.84*10^{-7}$ $d=1.07$	 $p=1.35*10^{-9}$ $d=1.63$	 $p=8.58*10^{-8}$ $d=1.25$
Without right hip (hipr; -7 features):	Without headcentre (-5 features):	
 $p=1.03*10^{-8}$ $d=1.07$	 $p=3.34*10^{-10}$ $d=1.48$	

We do see that the BFA is still capable of detecting treatment effects, even with clusterings based on a lower number of features. This goes in line with the results based on the B-SOiD and VAME clusterings (Supplementary Figure 2-4), which also run on a lower number of tracking points (# tracking points: B-SOiD = 9, VAME = 11).

Together with our studies on other hyperparameters (i.e. different clustering algorithms, number of clusters and integration periods), we do show that our approach is robust in detecting treatment effects. However, we do recognize that these studies are limited and that the influence of these choices depends on many other factors (for instance, a phenotype characterized by a change in gait may be much more influenced by a reliable tracking of leg movement than a phenotype that is characterized by decreased movement in general). We therefore included a section in the Discussion (page 13) to make the reader aware of the hyperparameter choices and that they may influence the results to different degrees.

- UMAP: Another reviewer raised a great point concerning UMAP (and its limitations). I took a closer look and agree with the points raised by the other reviewer. Additionally, UMAP itself has a lot of parameters and the embeddings are highly sensitive to these choices (ex: <https://umap-learn.readthedocs.io/en/latest/parameters.html>). The authors mentioned in the response they no longer use UMAP, but I still see a lot references of BFF with UMAP in the manuscript. Not sure if I've mis-understood the author response on this, or whether the paper has not been updated yet.

RE: We think this is just a misunderstanding, which we want to clarify here: Yes, we do still use UMAP to embed animals based on their behavior profile (transitions between clusters) into 2D. However, this is only done for visual comparison of treatments inside and across experiments. The use of UMAP embeddings to stratify animals into responders and non-responders, which was justifiably criticized previously by other reviewers, was replaced by the behavior flow likeness (BFL) score that does not rely on any UMAP embeddings.

- In the newest revision, the paper introduces several new terms (BFA, BFF, BFL) that could be confusing for readers and do not clearly convey the underlying methodology. More descriptive names would improve clarity and understanding. For example: BFA = "Manhattan distance-based bootstrapping" or a similar phrase that highlights the statistical methods used. BFF = "Dimensionality reduction of transition matrices". BFL = "difference from median". Additionally, the term "flow" could be misleading, as it implies a continuous and smooth transition. However, the method relies on discrete clusters of behaviors, and the transitions between these clusters might not always be smooth or continuous.

RE: We understand the reviewer's concern about terminology and the potential for acronyms to be confusing. However, we believe that transition between discrete clusters still captures the dynamics of animal behavior over time, thus the term 'behavioral flow' seems justified. The term has also been used in the referenced literature. We will check with the editors whether we should change this terminology. Over the years, we noticed in our own lab that it was extremely helpful to colloquially differentiate between the different analyses when analyzing complex experiments. In the text, we keep the use of acronyms to a minimum, but we think the nomenclature BF (behavioral flow) + one additional letter is rather intuitive. The basic bootstrapping Analysis is BFA, the Fingerprinting approach (BFF) assigns each mouse a single datapoint in high-dimensional space, and the assessment of Likeness (BFL) compares each datapoint to the median of other groups to see which group it resembles more closely.

To summarize the above points, my main concern is whether the proposed framework adds enough benefit (on top of the need to run clustering approaches) to justify the increased complexity.

RE: We hope to have addressed the remaining concerns raised by the reviewer.